# SPF45/RBM17-dependent, but not U2AF-dependent, splicing in a distinct subset of human short introns

Kazuhiro Fukumura [1✉], Rei Yoshimoto [1,2], Luca Sperotto[3,4], Hyun-Seo Kang [3,4], Tetsuro Hirose[5], Kunio Inoue[6], Michael Sattler[3,4] & Akila Mayeda [1✉]

Human pre-mRNA introns vary in size from under fifty to over a million nucleotides. We searched for essential factors involved in the splicing of human short introns by screening siRNAs against 154 human nuclear proteins. The splicing activity was assayed with a model HNRNPH1 pre-mRNA containing short 56-nucleotide intron. We identify a known alternative splicing regulator SPF45 (RBM17) as a constitutive splicing factor that is required to splice out this 56-nt intron. Whole-transcriptome sequencing of SPF45-deficient cells reveals that SPF45 is essential in the efficient splicing of many short introns. To initiate the spliceosome assembly on a short intron with the truncated poly-pyrimidine tract, the U2AF-homology motif (UHM) of SPF45 competes out that of U2AF$^{65}$ (U2AF2) for binding to the UHM-ligand motif (ULM) of the U2 snRNP protein SF3b155 (SF3B1). We propose that splicing in a distinct subset of human short introns depends on SPF45 but not U2AF heterodimer.

---

[1] Division of Gene Expression Mechanism, Institute for Comprehensive Medical Science, Fujita Health University, Toyoake, Aichi, Japan. [2] Department of Applied Biological Sciences, Faculty of Agriculture, Setsunan University, Hirakata, Osaka, Japan. [3] Institute of Structural Biology, Helmholtz Zentrum München, Neuherberg, Germany. [4] Bavarian NMR Center (BNMRZ), Chemistry Department, Technical University of Munich, Garching, Germany. [5] Graduate School of Frontier Biosciences, Osaka University, Suita, Japan. [6] Department of Biology, Graduate School of Science, Kobe University, Kobe, Hyogo, Japan. ✉email: fukumura@fujita-hu.ac.jp; mayeda@fujita-hu.ac.jp

There is a remarkable pattern in the distribution of higher eukaryotic pre-mRNA intron length; most introns fall either within a narrow peak under one hundred nucleotides or in a broad distribution peaking around several thousand nucleotides and extending to over a million nucleotides[1–3]. Pre-mRNA splicing is dependent upon a set of signal RNA elements recognized by essential factors that is a ubiquitous and essential part of eukaryotic gene expression. However, our understanding about specific and distinct mechanisms for the precise recognition of degenerated 5′ and 3′ splice site sequences within such extensively varied length of introns is fairly limited.

The canonical splicing mechanisms were studied and established using model pre-mRNAs with a single relatively short intron of a few hundred nucleotides, which are efficiently spliced in cells and in vitro[4,5]. According to such optimal systems, the essential splicing sequences in pre-mRNA, namely the 5′ splice site, the branch-site sequence, and the poly-pyrimidine tract (PPT) followed by the 3′ splice site, are initially recognized by the U1 snRNP, SF1, and the U2AF heterodimer (U2AF65/U2AF35, U2AF2/U2AF1 as HGNC approved symbol), respectively. Following the assembly of this early spliceosomal E complex, SF1 is replaced by the U2 snRNP in the A complex, which commits the intron for splicing reaction (reviewed in ref. [6]). The A complex is an asymmetric globular particle ($\sim 26 \times 20 \times 19.5$ nm)[7] that fully occupies 79–125 nucleotides (nt) of RNA[8], and recent high-resolution cryo-electron microscopy structures of the A complex have revealed molecular details of the overall architecture (reviewed in ref. [9]). Interestingly, human ultrashort introns with much shorter lengths (43–65 nt) are nevertheless spliced[10,11]. This raises the question of how such ultrashort introns can be recognized and committed to splicing by an 'oversized' A complex without steric hindrance.

We postulate that splicing of short introns depend on distinct specific factors, which utilize alternative ways for early spliceosomal assembly. Here, we have shown that this is the case in a subset of human short introns with the truncated PPT, which is recognized by a distinct constitutive splicing factor SPF45, but not by the authentic U2AF heterodimer.

## Results

**SPF45 is a specific essential splicing factor for a subset of short introns.** To find potential factors involved in splicing of short introns, we screened an siRNA library targeting 154 human nuclear proteins (including many known RNA-binding proteins and splicing factors) for splicing activity of the HNRNPH1 pre-mRNA including 56-nt intron 7[10,11] (Fig. 1a; Supplementary Table S1).

HeLa cells were transfected with each siRNA and recovered total RNAs were analyzed by RT-PCR to examine splicing activity of the endogenous HNRNPH1 pre-mRNA containing a 56-nt intron (Fig. 1a). The strongest splicing repression was markedly caused by knockdown of SPF45 (RBM17 as HGNC approved symbol; Fig. 1b) that indeed effectively depleted SPF45 protein (Fig. 1b, left panel; Supplementary Table S1). We also confirmed that knockdown of the best seven factors including SPF45, showing significant splicing repression in this short 56-nt intron (Supplementary Table S1, PSI > 0.3), but no repression at all on pre-mRNA splicing of control 366-nt intron (Supplementary Fig. S1a). To test if SPF45 might have a general role in splicing of short introns, we assayed two other endogenous pre-mRNAs targeting the 70-nt intron 9 of *RFC4* and the 71-nt intron 17 of *EML3*. The splicing efficiencies of these pre-mRNAs were also significantly repressed in SPF45-depleted HeLa cells (Fig. 1b).

Splicing inhibition was proportional to SPF45-knockdown efficiency induced by independent siRNAs (Supplementary Fig. S1b). These SPF45 siRNA-induced splicing defects were also observed in HEK293 cells, testifying to the robustness of our results (Supplementary Fig. S1c).

To test our hypothesis that SPF45 is indispensable to splice out short introns, we performed whole-transcriptome sequencing (RNA-Seq) with RNA from the SPF45-deficient HEK293 cells. The sequencing reads were mapped to the

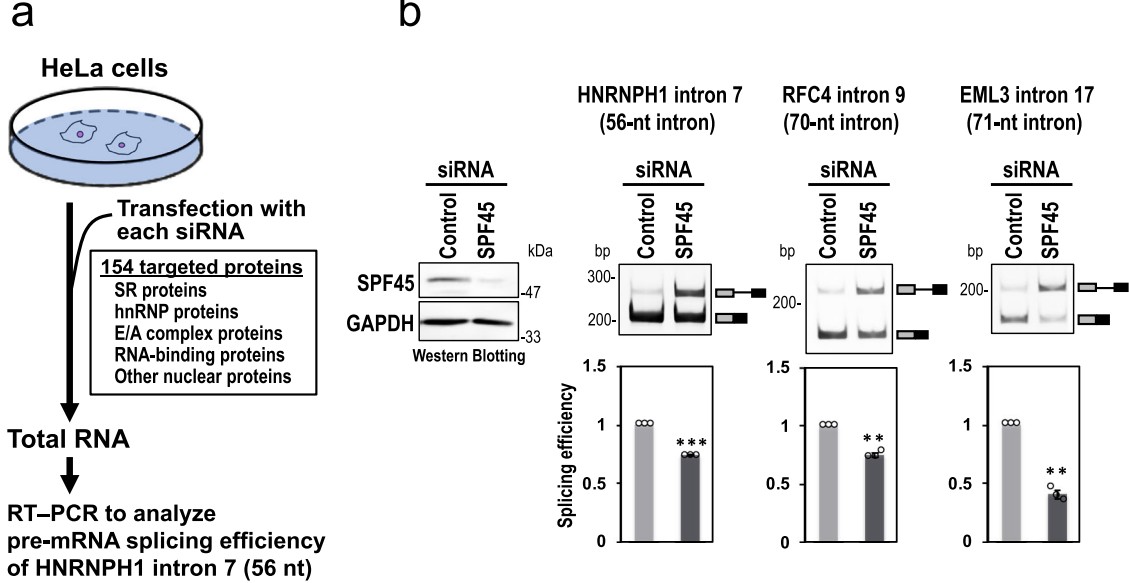

**Fig. 1 SPF45 was identified by siRNA screening for HNRNPH1 pre-mRNA (with 56-nt intron) splicing. a** The siRNA screening procedure to search for a specific splicing factor for short introns. **b** The SPF45 protein depletion by siRNA-knockdown, using SPF45 #2 siRNA (see Supplementary Fig. S1b), in HeLa cells was checked by a Western blotting (left panel). After the siRNA transfection, endogenous splicing of the indicated three representative short introns were analyzed by RT-PCR (right 3 panels). Means ± SEM are given for three independent experiments and two-tailed Student *t*-test values were calculated (HNRNPH1 intron: *p* = 0.0001 for Control vs SPF45 siRNA; RFC4 intron: *p* = 0.0035 for Control vs SPF45 siRNA; EML3 intron: *p* = 0.0031 for Control vs SPF45 siRNA). **P < 0.01, ***P < 0.001. Source data of the above panels are provided as a Source Data file.

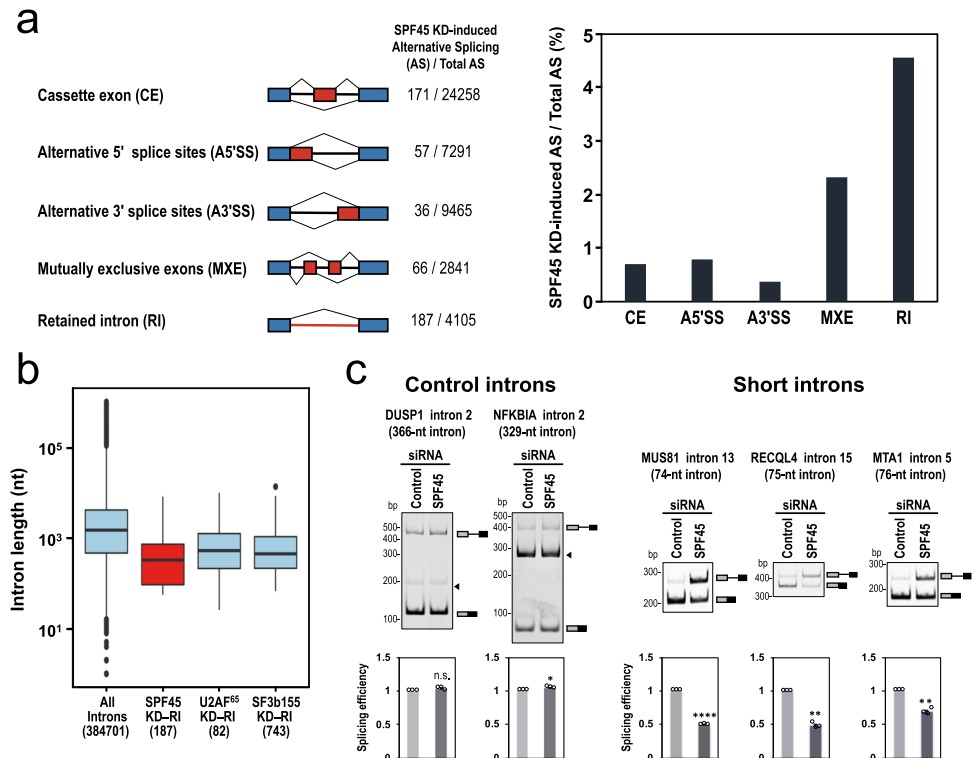

**Fig. 2 SPF45 is generally required for splicing of pre-mRNAs including short introns. a** RNA-Seq data exhibit deferential splicing patterns between control siRNA- and SPF45 siRNA-treated HEK293 cells. The numbers of significant splicing changes and total splicing events are indicated and the ratios are shown on the right. **b** Boxplots are comparing the intron-length distributions of all introns in human RefGene with those of the retained introns in SPF45-knockdown HEK293 cells. The retained introns in U2AF65- and SF3b155-knockdown HeLa cells obtained from the RNA-Seq data in GEO database are shown for comparison (significant for all pairs, $P < 0.05$). The numbers of introns are indicated in parentheses. Boxplots show the summary of the dataset; median (middle line), 25–75th percentile (box), and <5th and >95th percentile (whiskers), together with outliers (single points). **c** SPF45-knockdown selectively repressed splicing of pre-mRNAs with short introns. After the siRNA transfection in HEK293 cells, endogenous splicing of the indicated two control introns and three short introns were analyzed by RT-PCR. Arrowheads indicate nonspecific PCR products. Means ± SEM are given for three independent experiments and two-tailed Student t-test values were calculated (DUSP1 intron: $p = 0.0951$ for Control vs SPF45 siRNA; NFKBIA intron: $p = 0.0236$ for Control vs SPF45 siRNA; MUS81 intron: $p = 0.00004$ for Control vs SPF45 siRNA; RECQL4 intron: $p = 0.0019$ for Control vs SPF45 siRNA; MTA intron: $p = 0.0067$ for Control vs SPF45 siRNA). *$P < 0.05$, **$P < 0.01$, ****$P < 0.0001$, n.s.(not statistically significant) $P > 0.05$. Source data of all the above panels are provided as a Source Data file.

human genome reference sequence. We identified 517 changes in splicing from a total of 47,960 alternative splicing events (Fig. 2a, left panel). The most frequent changes of splicing in SPF45-depleted HEK293 cells were intron retention events (Fig. 2a, right graph; see Supplementary Data 1 for the list of all 187 introns).

The analysis of these retained introns hinted at a potential mechanism for the role of SPF45. Remarkably, the length distribution of the retained introns in SPF45-depleted cells is strongly biased towards shorter lengths compared to those in cells depleted of constitutive splicing factors, U2AF65 and SF3b155 (SF3B1 as HGNC approved symbols), which show a distribution comparable to the whole set of introns (Fig. 2b).

We validated these RNA-Seq-based profiles by RT-PCR. As assumed, splicing of pre-mRNAs with two control introns (366 and 329 nt) were not affected by SPF45 knockdown, while in contrast, three arbitrarily chosen pre-mRNAs with short introns (74, 75, and 76 nt) were repressed (Fig. 2c). These results demonstrate that SPF45 is required for the efficient splicing of a substantial population of pre-mRNAs with short introns.

**SPF45 is required for splicing on intron with truncated poly-pyrimidine tract (PPT).** Next we searched for a potential cis-element in short introns through which SPF45 might act. From

RNA-Seq data of SPF45-depleted cells, we found that strengths of the 5′/3′ splice sites and the branch sites of SPF45-dependent short introns are somewhat weaker than the average in RefGene (Supplementary Fig. S2a). Therefore, we first examined these cis-acting splicing signals using mini-gene splicing assays in SPF45-depleted cells. As expected, splicing of HNRNPH1 pre-mRNA (56-nt intron 7; use in our siRNA screening) was repressed by depletion of SPF45, whereas splicing of the control adenovirus 2 major late (AdML) pre-mRNA (231-nt intron 1; used as a standard splicing substrate previously) was unaffected (Fig. 3; top-row). The SPF45-dependent splicing of the HNRNPH1 pre-mRNA was not altered even after replacement of the 5′/3′ splice sites and the branch site individually, or all together, by those of the AdML pre-mRNA (Supplementary Fig. S2b). These results indicate that the requirement of SPF45 depend on neither the 5′/3′ splice sites nor the branch site.

We then examined whether the SPF45 dependency is attributed to the PPT. The PPT score (see Methods) is one of the criteria to evaluate effective PPTs: PPT scores are 19 for the PPT (13 nt) in HNRNPH1-intron 7 and 52 for the PPT (25 nt) in AdML-intron 1 (Fig. 3, second-row). Remarkably, the SPF45-dependent splicing in HNRNPH1 pre-mRNA was altered toward the SPF45-independent splicing by replacement of the HNRNPH1-PPT with the conventional AdML-PPT (Fig. 3, AdML PPT25). This alteration was not due to the extended

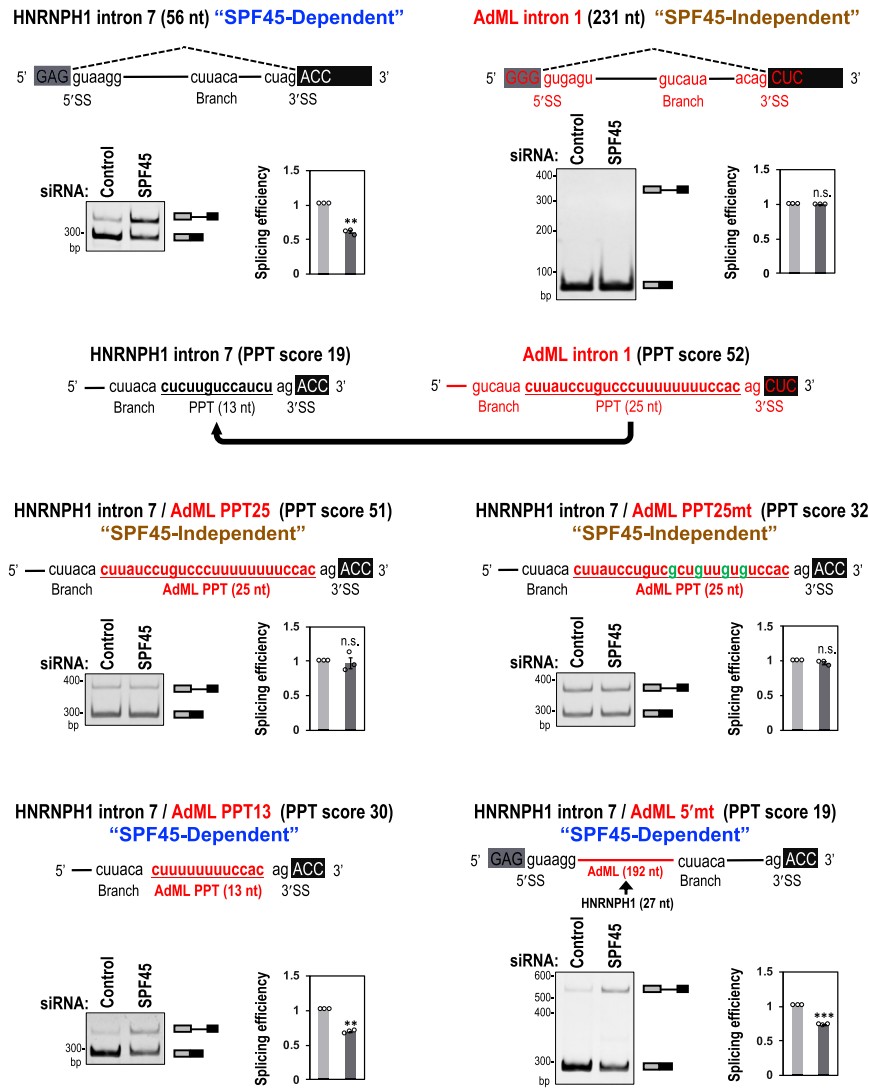

**Fig. 3 The length of poly-pyrimidine tracts (PPTs) determine the SPF45 dependency of splicing.** Original HNRNH1 and AdML pre-mRNAs and chimeric HNRNH1 pre-mRNAs are schematically shown (red color indicates AdML derived sequences). These pre-mRNAs were expressed from mini genes in HeLa cells and their splicing was assayed by RT-PCR. PAGE images and quantifications of RT-PCR are shown. Means ± SEM are given for three independent experiments and two-tailed Student $t$-test values were calculated ($p = 0.0027$ for Control siRNA vs SPF45 siRNA in HNRNPH1 intron, $p = 0.9394$ for Control vs SPF45 siRNA in AdML intron, $p = 0.7548$ for Control siRNA vs SPF45 siRNA in HNRNPH1 intron/AdML PPT25, $p = 0.1961$ for Control siRNA vs SPF45 siRNA in HNRNPH1 intron/AdML PPT25mt, $p = 0.0011$ for Control siRNA vs SPF45 siRNA in HNRNPH1 intron/AdML PPT13, $p = 0.0004$ for Control siRNA vs SPF45 siRNA in HNRNPH1 intron/AdML 5′mt. **$P < 0.01$, ***$P < 0.001$, n.s.$P > 0.05$. Source data of the above panels are provided as a Source Data file.

distance between the branch site and the 3′ splice site in 'AdML PPT25' since the same extended distance in SPF45-dependent pre-mRNAs did not lose the SPF45 dependency (Supplementary Fig. S3a, +12nt and AdML PPT13/+12nt). To determine whether SPF45 recognizes the strength or the length of a given PPT, we reduced the PPT score of AdML-PPT in two ways: one was transversion mutations in the PPT (C/U→G; score 52→32), and the other was truncation of the PPT (25 nt→13 nt; score 52→30). Notably, the transversion mutations in the PPT did not gain SPF45 dependency (Fig. 3, AdML PPT25mt) but the truncation of PPT did (Fig. 3, AdML PPT13).

Lastly, we expanded the distance between the 5′ splice site and the branch site in HNRNPH1 intron (27 nt) by replacement with the corresponding fragment in the AdML intron (192 nt). Interestingly, this chimeric pre-mRNA with the short HNRNPH1 PPT remained SPF45 dependent (Fig. 3, AdML 5′mt). Taken together, these results demonstrate that short PPT per se in the

HNRNPH1-intron 7 is the determinant for the SPF45 dependency in splicing.

These observations were further recapitulated and validated in the distinct SPF45-dependent EML3 pre-mRNA, which contains a 71-nt intron (Supplementary Fig. S3b). Moreover, our global PPT length analysis of the retained introns in SPF45-depleted cells showed that PPT lengths of SPF45-dependent short introns (<100 nt) were significantly shorter than those of the whole RefGene introns (Supplementary Fig. S4). Therefore, we can define introns shorter than ~100 nt as 'short intron' in this study. Together, it is the truncated PPTs in short introns that are crucial for the SPF45 function in splicing.

**SPF45 replaces U2AF$^{65}$ on truncated PPTs in short introns to promote splicing.** In an early transition of human spliceosome from the E to A complexes, the branch site, PPT and the 3′ splice

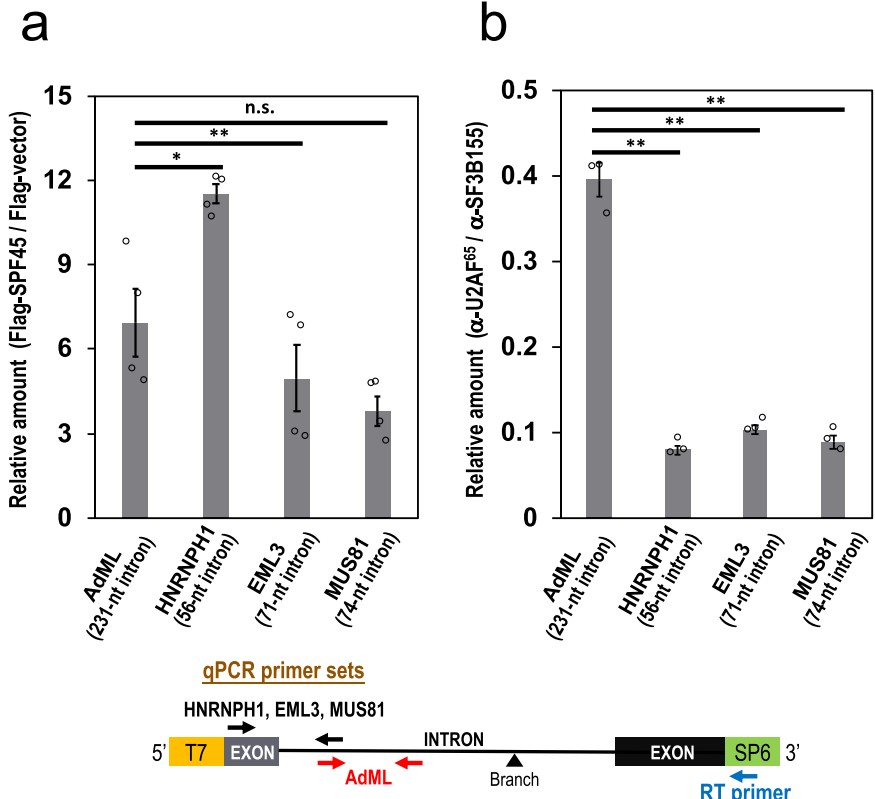

**Fig. 4 SPF45 binds to all introns while U2AF$^{65}$ cannot bind to short introns. a** Cellular formaldehyde crosslinking and immunoprecipitation experiments shows SPF45 binding to all the indicated introns. Mini genes containing these four introns were individually co-transfected into HEK293 cells with a plasmid expressing Flag-SPF45 protein. The Flag-SPF45 was immunoprecipitated after formaldehyde crosslinking and then co-precipitated pre-mRNAs were quantified by RT–qPCR using specific primers (see the schematic mini gene below). Means ± SEM are given for three independent experiments and two-tailed Student $t$-test values were calculated ($p = 0.0353$ for AdML vs HNRNPH1 intron, $p = 0.0083$ for AdML vs EML3 intron, $p = 0.0638$ for AdML vs MUS81 intron). *$P < 0.05$, ***$P < 0.0005$, n.s. $P > 0.05$. **b** Cellular CLIP experiments shows strong U2AF$^{65}$ binding to control AdML pre-mRNA but not much to the three indicated short introns. Mini genes containing these four introns were individually co-transfected into HEK293 cells and irradiated with UV light. The lysates were immunoprecipitated with anti-U2AF$^{65}$ and anti-SF3b155 antibodies and then immunoprecipitated RNAs were quantified by RT–qPCR using specific primers (see the schematic mini gene below). Means ± SEM are given for three independent experiments and two-tailed Student $t$-test values were calculated ($p = 0.0034$ for AdML vs HNRNPH1 intron, $p = 0.0066$ for AdML vs EML3 intron, $p = 0.0020$ for AdML vs MUS81 intron). **$P < 0.01$. Source data of the above graphs are provided as a Source Data file.

site are bound cooperatively by SF1, U2AF$^{65}$, and U2AF$^{35}$, respectively. A stable U2 snRNP-associated splicing complex is then formed by ATP-dependent displacement of SF1, where the p14 protein (a U2 snRNP-associated factor) contacts the branch site, and U2AF$^{65}$ interacts with the U2 snRNP component SF3b155 (reviewed in ref. [6]). Since the splicing activity of SPF45 depends on short PPTs, we hypothesized that insufficient U2AF$^{65}$ binding to a truncated PPT would allow its replacement by SPF45.

To examine whether SPF45 associates with short introns, mini genes encoding HNRNPH1, EML3, MUS81, and control AdML introns, were expressed in HeLa cells. We confirmed that these mini genes were spliced in an SPF45-dependent manner (Fig. 3 and Supplementary Fig. S5) as we observed in these endogenous genes (Figs. 1 and 2c), so we proceeded to analyze proteins associated with these ectopically expressed RNAs.

We used formaldehyde crosslinking to detect any indirect RNA association of SPF45[12]. HeLa cells were co-transfected with the four mini genes and Flag-SPF45 expression plasmid, crosslinked with formaldehyde, immunoprecipitated with anti-Flag antibody, and the co-precipitated RNAs were analyzed by RT-qPCR (Fig. 4a). SPF45 associated to all pre-mRNAs derived from these four mini genes irrespective of the length of the introns. The association of SPF45 to control AdML intron (231 nt) is

consistent with previous proteomic detection of SPF45 in the human spliceosome[13]. SPF45 was also detected on MINX (131 nt) and PM5 (235 nt) introns[14,15] (reviewed in ref. [6]).

We next examined the binding of U2AF$^{65}$ and SF3b155 to pre-mRNAs by UV crosslinking–immunoprecipitation (CLIP). Whole-cell extracts from crosslinked cells were immunoprecipitated with anti-U2AF$^{65}$ and anti-SF3b155 antibodies, and the precipitated RNAs were analyzed by RT-qPCR (Fig. 4b). SF3b155, as a component of the U2 snRNP, bound to all the pre-mRNAs, while significant U2AF$^{65}$ binding was observed only with control AdML pre-mRNA and it was very weak with the three SPF45-dependent short introns.

We biochemically verified the binding of U2AF$^{65}$ and SF3b155 to the HNRNPH1 intron and control AdML intron by affinity pull-down assay. The biotinylated HNRNPH1 and AdML pre-mRNAs were transcribed in vitro, incubated with HEK293 cell nuclear extracts, and interacted proteins were examined by Western blotting (Fig. 5). In agreement with our formaldehyde crosslinking experiments, SPF45 associated with both AdML and HNRNPH1 pre-mRNAs (see Discussion). However, importantly, U2AF$^{65}$ strongly bound only to AdML pre-mRNA but U2AF$^{65}$ was allowed to associate HNRNPH1 pre-mRNA only if SPF45 was depleted from nuclear extracts. These results together support our proposed hypothesis that SPF45 replaces U2AF$^{65}$

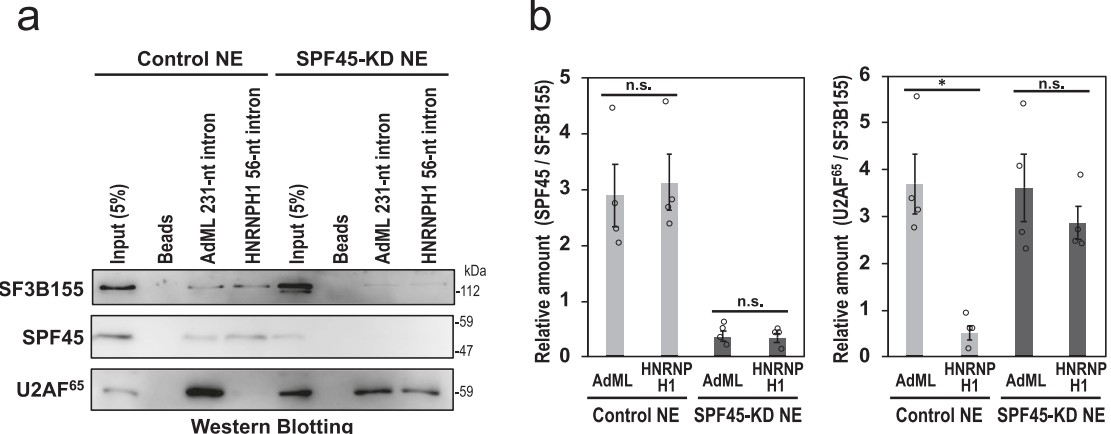

**Fig. 5 Binding of SPF45 competes out U2AF65 on truncated PPTs in short introns. a** Affinity pull-down experiments of biotinylated RNA indicates U2AF65 binding to the short intron only if SPF45 was depleted. Biotinylated pre-mRNAs including short HNRNPH1 intron (56 nt) and control AdML intron (231 nt) were incubated with nuclear extracts from either control siRNA- or SPF45 siRNA-treated HEK293 cells. The biotinylated RNA-bound proteins were pulled down with streptavidin-coated beads and analyzed by Western blotting with antibodies against SF3b155, SPF45, and U2AF65. **b** The graph shows quantification of the bands on Western blots (panel **a** as the representative blots). SPF45 and U2AF65 scanned data were normalized to SF3B155 (SPF45/SF3B155 and U2AF65/SF3B155) and plotted. Means ± SEM are given for four independent experiments and two-tailed Student t-test values were calculated (Left panel: $p = 0.2982$ for AdML vs HNRNPH1 intron in Control NE; $p = 0.6973$ for AdML vs HNRNPH1 intron in SPF45-KD NE; Right panel: $p = 0.0142$ for AdML vs HNRNPH1 intron in Control NE, $p = 0.4647$ for AdML vs HNRNPH1 intron in SPF45-KD NE). *$P < 0.05$, n.s. $P > 0.05$. Source data of all the above panels are provided as a Source Data file.

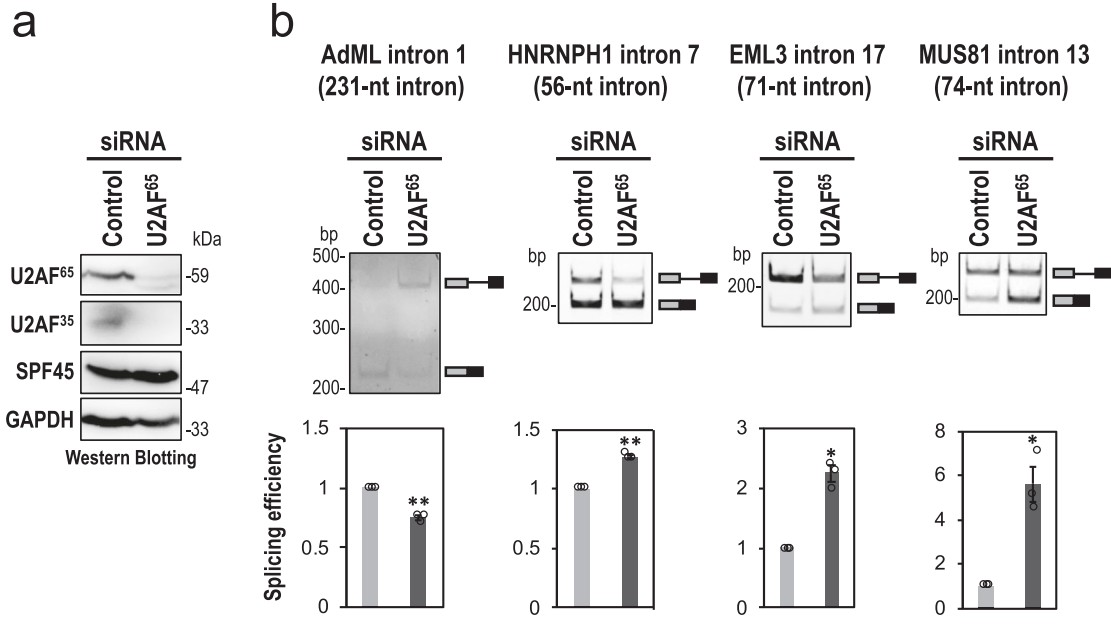

**Fig. 6 Depletion of U2AF65 rather increases splicing of pre-mRNAs with SPF45-dependent short introns. a** The co-depletion of U2AF65 and U2AF35 proteins from HeLa cells was observed by a Western blotting. Displayed blots are representative of three independent experiments. **b** After the siRNA transfection, HeLa cells were cultured for short time (4 h) to effectively observe splicing stimulation, and splicing efficiencies of the indicated four mini genes were analyzed by RT-PCR. Means ± SEM are given for three independent experiments and two-tailed Student t-test values were calculated (AdML intron: $p = 0.0067$ for Control vs U2AF65 siRNA; HNRNPH1 intron: $p = 0.0038$ for Control vs U2AF65 siRNA; EML3 intron: $p = 0.0116$ for Control vs U2AF65 siRNA; MUS81 intron: $p = 0.0273$ for Control vs U2AF65 siRNA. *$P < 0.05$, **$P < 0.01$. Source data of all the above panels are provided as a Source Data file.

in the assembly of U2 snRNP complexes as U2AF65 is poorly bound to truncated PPTs of short introns.

We noticed that endogenous U2AF65 knockdown barely repressed splicing of the SPF45-dependent short intron (Supplementary Table S1, No. 142; Supplementary Fig. S6a). Therefore, we checked splicing efficiencies of these four pre-mRNAs in U2AF65-knockdown HeLa cells (Fig. 6). This depletion of

U2AF65 also caused effective co-depletion of U2AF35 (Fig. 6a) that is consistent with previous reports[16,17]. In the control AdML pre-mRNA, spliced mRNA was reduced by the depletion of U2AF65, showing that U2AF heterodimer is essential for conventional AdML pre-mRNA splicing as expected. Remarkably, splicing of SPF45-dependent pre-mRNAs with short introns was rather activated by the depletion of U2AF65 (Fig. 6b).

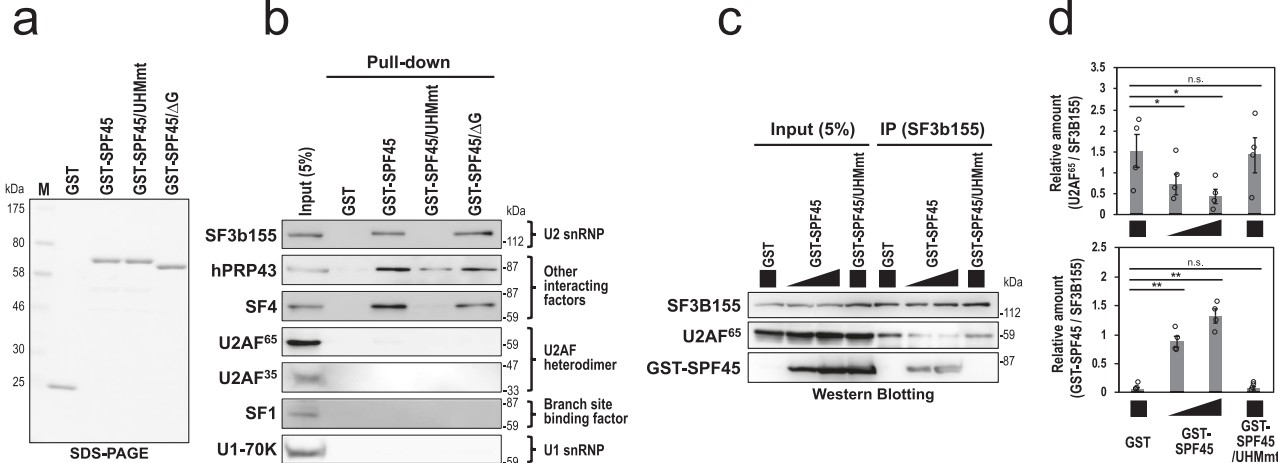

**Fig. 7 ULM–UHM binding between SF3b155 and SPF45 promotes splicing of pre-mRNA with short introns. a** SDS-PAGE analysis of the indicated purified recombinant proteins. **b** GST pull-down assay in RNase A-treated HeLa nuclear extract shows the UHM-dependent binding of GST-SPF45 to SF3b155 but not to U2AF heterodimer. Proteins that associated with these GST-fusion proteins were detected by Western blotting using the indicated antibodies. **c** Immunoprecipitation of SF3b155 with the indicated GST-fusion proteins shows the competitive binding between U2AF[65] and SPF45 to SF3b155 via UHM–ULM interactions. The same reaction mixtures in panel **a** were immunoprecipitated with anti-SF3b155 antibody and the associated proteins were analyzed by Western blotting with antibodies against SF3b155, U2AF[65], and SPF45. **d** The graph shows quantification of the bands on Western blots (panel **b** as the representative blots). U2AF[65] and GST-SPF45 scanned data were normalized to SF3B155 (U2AF[65]/SF3B155 and GST-SPF45/SF3B155) and plotted. Means ± SEM are given for four independent experiments and two-tailed Student *t*-test values were calculated (Upper panel: p = 0.0470 for GST vs GST-SPF45 left, p = 0.0322 for GST vs GST-SPF45 right, p = 0.6238 for GST vs GST-SPF45/UHMmt; Lower panel: p = 0.0036 for GST vs GST-SPF45 left, p = 0.0010 for GST vs GST-SPF45 right, p = 0.5023 for GST vs GST-SPF45/UHMmt). *P < 0.05, **P < 0.01, n.s.P > 0.05. Source data of all the above panels are provided as a Source Data file.

In endogenous SPF45-dependent pre-mRNAs (Supplementary Fig. S6a), such marked activation was not observed that could be due to the almost saturated efficiency of splicing (see amounts of unspliced pre-mRNAs in Supplementary Fig. S6a). Taken together, we conclude that SPF45 effectively competes out U2AF heterodimer on truncated PPTs and the newly installed SPF45 promotes splicing of pre-mRNAs with short introns.

**SF3b155–U2AF[65]/U2AF[35] is displaced by SF3b155–SPF45 via ULM–UHM binding.** The SPF45 protein contains a G-patch motif that may interact with nucleic acids and proteins[18,19], and a C-terminal U2AF-homology motif (UHM) that binds the UHM-Ligand motifs (ULM) of its partner proteins. UHM–ULM interactions; e.g., U2AF[65](UHM)–SF1(ULM), U2AF[65](UHM)–SF3b155 (ULM), and U2AF[35](UHM)–U2AF[65](ULM), plays an essential role in the splicing reactions[20–22] (reviewed in ref. [23]). Remarkably, in vitro binding analyses using the purified recombinant proteins showed that the UHM of SPF45 can bind to the ULMs of SF3b155, U2AF[65], and SF1; on the other hand, the UHM and G-patch motif of SPF45 cannot bind directly to RNA[22]. We therefore postulated that the SF3b155–U2AF[65]/U2AF[35] complex is remodeled to the SF3b155–SPF45 complex by switching of their ULM–UHM interactions and that SPF45 per se does not necessarily bind to the truncated PPT.

To test our hypothesis, we first examined the binding of SPF45 to SF3b155. We prepared *E. coli* recombinant glutathione S-transferase (GST)-fusion proteins of SPF45 and its variants (Fig. 7a). The D319K mutant in the UHM (SPF45/UHMmt) no longer binds any ULM, and G-patch motif-deleted mutant (SPF45/ΔG) loses potential interaction with nucleic acids and proteins[22] (Supplementary Fig. S7a). Although both SF3b155 and SF1 contain ULM and they can interact with SPF45 in vitro[22], our GST pull-down assays with crude nuclear extracts, closer to physiological conditions, demonstrated that GST-SPF45 bound to SF3b155, but not to SF1 (Fig. 7b). As expected, the UHM of SPF45 is essential (Fig. 7b, GST-SPF45/UHMmt) while the G

patch of SPF45 is dispensable (GST-SPF45/ΔG), confirming the ULM–UHM interaction in the SF3b155–SPF45 complex. GST-SPF45 also binds to two other previously suggested SPF45-partner proteins that lack ULMs: spliceosomal A-complex protein, SF4 (SUGP1 as HGNC approved symbol), and DEAH helicase protein of the U2-related group, hPRP43[24]. However, we confirmed that these two SPF45-interacting factors are not relevant to splicing of short introns (Supplementary Fig. S6b, c).

Using formaldehyde crosslinking assay, we also confirmed the importance of the ULM–UHM interaction in the recruitment of SPF45 to the target short introns (Supplementary Fig. S8). The SPF45-UHM mutant (Flag-SPF45/UHMmt/siR) showed the remarkable impairment of the association to short introns, indicating that the recruitment of SPF45 to the truncated PPT is dependent on SF3b155 binding through ULM–UHM interaction. Using NMR analysis, we tested whether SPF45 can bind to truncated PPT RNA of the SPF45-dependent short introns in vitro. Our NMR titration experiments indicate that neither the G-patch motif nor the UHM domain of SPF45 shows significant affinity towards these two truncated PPT RNAs (Supplementary Fig. S9). Taken together, the recruitment of SPF45 to the target short introns may not depend on its direct binding to truncated PPT, but rather it requires the protein interaction with SF3b155 (Fig. 9).

The binding between U2AF[65](ULM) and U2AF[35](UHM) is very strong[20]. Remarkably, GST-SPF45 did not pull-down U2AF[65] and U2AF[35] in crude nuclear extracts (Fig. 7b). These data together suggest that the U2AF[65](ULM) does not interact with the SPF45(UHM) in nuclear extracts, so that the SPF45 (UHM) and U2AF[65](UHM) compete for a functional binding toward the SF3b155(ULM) (Fig. 9). Therefore, we next investigated the competitive binding of U2AF[65] and SPF45 toward SF3b155 by titrating the dose of GST-SPF45 in the immunoprecipitation assays (Fig. 7c, d). Notably, GST-SPF45 interfered with the binding between SF3b155 and U2AF[65] in a dose-dependent manner; however, GST-SPF45/UHMmt did not

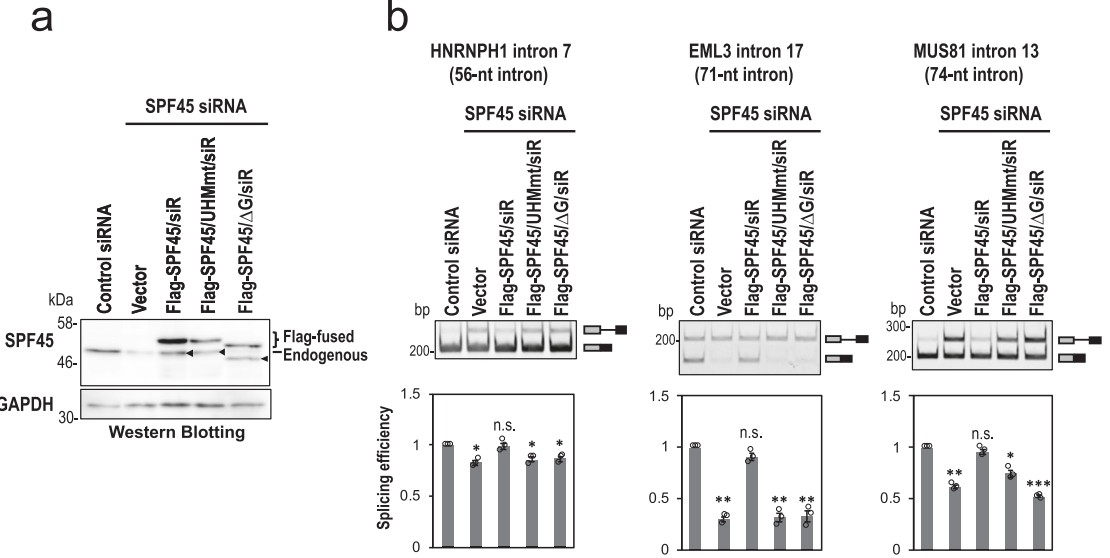

**Fig. 8 The expression of siRNA-resistant SPF45 proteins rescue splicing of short introns in SPF45-depleted cells. a** The expressed Flag-fused siRNA-resistant (siR) proteins and endogenous SPF45 in HeLa cells were checked by Western blotting. Displayed blots are representative of three independent experiments. Arrowheads indicate degraded siRNA-resistant (siR) proteins, but not endogenous SPF45 (see 'vector' lane for the depletion efficiency of endogenous SPF45). **b** After the co-transfection of the indicated siRNA-resistant plasmids and three mini-gene plasmids, splicing efficiencies of the indicated three mini genes were analyzed by RT-PCR. Means ± SEM are given for three independent experiments and two-tailed Student $t$-test values were calculated (HNRNPH1 intron: $p = 0.0175$ for Control vs Vector, $p = 0.8177$ for Control vs Flag-SPF45/siR, $p = 0.0270$ for Control vs Flag-SPF45/UHMmt/siR, $p = 0.0250$ for Control vs Flag-SPF45/ΔG/siR; EML3 intron: $p = 0.0012$ for Control vs Vector, $p = 0.1085$ for Control vs Flag-SPF45/siR, $p = 0.0035$ for Control vs Flag-SPF45/UHMmt/siR, $p = 0.0057$ for Control vs Flag-SPF45/ΔG/siR; MUS81 intron: $p = 0.0020$ for Control vs Vector, $p = 0.1693$ for Control vs Flag-SPF45/siR, $p = 0.0177$ for Control vs Flag-SPF45/UHMmt/siR, $p = 0.0004$ for Control vs Flag-SPF45/ΔG/siR). *$P < 0.05$, **$P < 0.01$, ***$P < 0.001$, n.s.$P > 0.05$. Source data of all the above panels are provided as a Source Data file.

disturb this binding. These results indicate that the SPF45(UHM) competes with the U2AF65(UHM) for the SF3b155 binding.

Finally, we examined whether the SPF45–SF3b155 interaction and the G patch of SPF45 are essential for the SPF45-dependent splicing on short introns. We performed functional rescue experiments with SPF45-depleted HeLa cells using three siRNA-resistant proteins; SPF45 (SPF45/siR), SPF45-UHM mutant (SPF45/UHMmt/siR), and a G-patch motif-deleted SPF45 (SPF45/ΔG/siR; Supplementary Fig. S7a). We confirmed that the subcellular localization of these three mutant proteins did not change from that of endogenous SPF45 protein (Supplementary Fig. S7b). Protein expression levels of endogenous SPF45 and ectopically expressed three SPF45 siRNA-resistant mutants were checked by Western blotting in SPF45-depleted HeLa cells (Fig. 8a). We analyzed splicing efficiencies of three pre-mRNAs including short introns by RT-PCR (Fig. 8b). SPF45/siR rescued the splicing defects of all short introns in SPF45-depleted HeLa cells, however, the SPF45/UHMmt/siR and SPF45/ΔG/siR did not (compare with control 'Vector' lane). Taken together, we conclude that it is SPF45 that competes out U2AF65 and SPF45 is localized at the truncated PPT via protein–protein interaction with the U2 snRNP component SF3b155 to promote splicing of pre-mRNAs with short introns.

## Discussion

Over a generation ago, two different splicing mechanisms termed 'intron-definition model' for short introns and 'exon-definition model' for long introns were proposed (reviewed in ref. [25]). In the former model, the frequent lack of a canonical PPT in vertebrate short introns was noticed and an alternative mechanism that circumvents this problem were postulated. Here we provide one of answers to this puzzling question by demonstrating that a subset of human short introns, with significantly undersized pyrimidine

tracts, is recognized by SPF45 but not by the authentic U2AF heterodimer; implicating that SPF45 is a distinct constitutive splicing factor in the spliceosomal complex A. This finding rationally answer the question of why SPF45, which was previously considered just to be an alternative splicing factor, is essential for cell survival and maintenance in vivo[26]. Since the conditional knockout of SPF45 in mice causes extensive dysregulation of splicing[26], it is reasonable to assume that SPF45-dependent splicing of pre-mRNAs including short introns with truncated PPT could be a part of the targets of such dysregulation.

**A mechanistic model of SPF45-dependent splicing.** We found that both SPF45 and U2AF65 can bind on introns with SF3b155 irrespective of intron size presumably via interactions with the five ULMs in SF3b155, as previously shown in the simultaneous binding of U2AF65 and PUF60 to SF3b155[27]. This observation is justified by the previous mass-spectrometry analysis using AdML, MINX, and PM5 pre-mRNAs with conventional introns; i.e., SPF45 is contained in E, A and B complexes as a U2 snRNP-related protein[13–15,28,29] (reviewed in ref. [30]).

It was demonstrated that SPF45-depleted fruit fly (*Drosophila melanogaster*) S2 cells can be functionally rescued by human SPF45[31]. The SPF45-dependent splicing event on the shorter intron might be conserved in fruit fly. Interestingly, fruit fly spliceosomal B complex formed on Zeste pre-mRNA (with 62-nt intron including 14-nt PPT) contains SPF45, but that formed on Ftz pre-mRNA (with 147-nt intron including 33-nt PPT) does not[32]. Therefore, SPF45 is located preferentially in short introns in fruit fly, while in human, SPF45 exists in all introns on standby mode for short introns.

Our considerable finding is that U2AF65 needs to be expelled by SPF45 to promote splicing of a subset of short introns. Structural studies showed that U2AF65 recognizes eight or nine

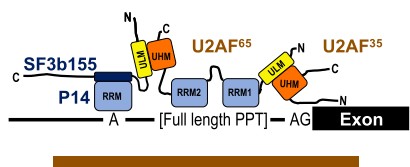

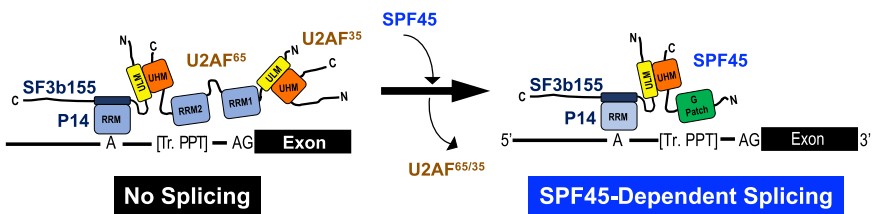

**Fig. 9 The model of U2AF-dependent splicing on a conventional-sized intron with a regular PPT and SPF45-dependent splicing on a short intron with a truncated PPT.** The associated splicing factors with the domain structures and the target sequences of pre-mRNAs are represented schematically. On short intron with truncated PPT (Tr. PPT), U2AF heterodimer is replaced by SPF45 and interacts with SF3b155 (U2 snRNP component) via UHM–ULM binding to promote splicing. Since both U2AF[65] and SPF45 can bind to long introns simultaneously via five ULMs in SF3b155, the alternative model that U2AF[65] displaces SPF45 in long introns would be impossible.

nucleotides of pyrimidine tract[33,34]. The high affinity RNA binding and efficient U2AF-dependent splicing requires at least eight pyrimidines[33], which are likely to be part of a rather extended PPT[35]. We thus propose a mechanistic model that the weak and unstable U2AF[65] binding on the truncated PPT of short intron triggers the protein interaction of SPF45 with SF3b155, leading to the structural and functional replacement of the U2AF heterodimer by SPF45 (Fig. 9).

Our NMR data revealed that SPF45, either the UHM domain alone or in the presence of the G patch, does not significantly bind the truncated PPT on its own in vitro, and our biochemical data showed protein interaction between SPF45 and SF3b155 is essential for SPF45 to be settled around the truncated PPT of short introns. However, our results clearly implicate the critical role of the G patch in SPF45-dependent splicing. We thus assume that SPF45 is recruit in the functional position of short introns by an interaction with the third protein factor, which could be involved in the recognition of 3′ splice sites. Our effort is underway to identify this SPF45-interacting factor.

Using the RNA-Seq approach, we identify 187 introns whose splicing is SPF45 dependent (Supplementary Data 1). The length distribution of SPF45-dependent introns was extensive (56–7699 nt) and a subset of the long introns do not necessarily possess truncated PPTs (statistically analyzed in Supplementary Fig. S4). The determinant and mechanism for SPF45-dependent splicing on the longer introns (>102 nt) are under investigation. Nevertheless, we could conclude that the set of SPF45-dependent short introns (<100 nt) possess the significantly shorter PPTs (Supplementary Fig. S4), and these are spliced out by our proposed distinct mechanism.

**Different SPF45-mediated mechanisms in constitutive splicing and alternative splicing.** Our RNA-Seq analysis in the SPF45-knockdown cells detected the changes in alternative splicing (Fig. 2a). SPF45 was indeed identified and characterized as an alternative splicing regulator. In fruit fly, SPF45 interacts with Sex lethal (Sxl) protein and induces exon 3 skipping of Sxl pre-mRNA[31]. In mammals, SPF45 can cause exon 6 skipping in FAS pre-mRNA[22] and it produces soluble isoform of FAS, which may contribute in the regulation of apoptosis (reviewed in ref. [36]).

In the SPF45-induced regulation of alternative splicing, there was no competition between SPF45 and U2AF heterodimer on the Sxl pre-mRNA[31]. Whereas we found a competitive and mutually exclusive binding of SPF45 and U2AF heterodimer on the truncated PPT to splice out short intron. We speculate that the cooperative interaction of SPF45 and U2AF[65] with SF3b155 may be required for alternative splicing regulation, whereas, exclusive binding of SPF45 with SF3b155, but without the U2AF heterodimer, is critical for short intron-specific constitutive splicing (Fig. 9).

The U2-related protein PUF60 and U2AF[65] cooperatively interact with SF3b155 to activate weak 3′ splice sites[37] and the potential binding of both proteins to SF3b155 was demonstrated[27]. Interestingly, cellular knockdown of SPF45, U2SURP, or CHERP caused the extensive alteration in annotated and unannotated alternative splicing, indicating functional interactions among these three factors in the regulation of alternative splicing[38]. We also observed that knockdown of SPF45 causes splicing alterations in various types (Fig. 2a). However, our siRNA screening showed that knockdown of PUF60 and U2SURP have no effect on the splicing of HNRNPH1 short intron (Supplementary Table S1). Consistently, the event that knockdown of SPF45 causes the inclusion of retained intron, which may reflect the repression of short introns, is not overlapped by knockdown of U2SURP or CHERP[38].

Together, we conclude that the critical mechanism of SPF45 as a constitutive splicing factor is distinct from the mechanism of SPF45, together with other interactors, as an alternative splicing factor.

**Another distinct subsets of human short introns.** Here, we have just described one distinct subset of human SPF45-dependent short introns. Most recently, Smu1 and RED proteins were shown to activate spliceosomal B complexes assembled on human short introns[39]. Notably, Smu1/RED are human-specific splicing factors, whereas SPF45 is evolutionarily conserved from fruit fly. The distance between the 5′ splice site and branch site needs to be sufficiently short for Smu1/RED-dependent splicing, whereas in contrast, we clearly showed that this distance per se is not responsible for SPF45-dependent splicing (Fig. 3, AdML 5′mt). Smu1/RED relieve physical constraints arising from this short

distance, so that spliceosomes can overcome structural constraint associated with short introns. To explore the structural constraint on SPF45-dependent short intron, we definitely need to analyze the spliceosomal complex A formation by mass-spectrometry and cryo-electron microscopy. In short, Smu1/Red-dependent and SPF45-dependent splicing mechanisms are essentially different, and thus they target two distinct subsets of human short introns.

The subset of SPF45-dependent short introns was identified by splicing activity of the HNRNPH1 pre-mRNA with 56-nt intron 7 that contains conventional splice sites and branch site. We previously validated a list of ultrashort introns that includes remarkably atypical G-rich introns with completely inefficient splice sites and branch sites, in which the 49-nt intron 12 of the NDOR1 gene and the 43-nt intron 6 of the ESRP2 gene were analyzed[10,11]. The mechanism of splicing involved in such atypical G-rich introns with inevitable steric hindrance is enigmatic. We assume the existence of another exotic subset of human ultrashort G-rich introns.

## Methods

**Construction of expression plasmids**. The mini-gene expression plasmids, pcDNA3-HNRNPH1, pcDNA3-EML3, and pcDNA3-MUS81, were constructed by subcloning the corresponding PCR-amplified fragment into pcDNA3 vector (Invitrogen–Thermo Fisher Scientific). The PCRs were performed using genomic DNA of HeLa cells and specific primer sets (Supplementary Table S2). For the pcDNA3-AdML, the PCR was performed using the pBS-Ad2 plasmid[40] and specific primer sets (Supplementary Table S2).

The chimeric expression plasmids, pcDNA3-HNRNPH1/5'SSAdML, pcDNA3-HNRNPH1/branchAdML, pcDNA3-HNRNPH1/3'SSAdML, pcDNA3-HNRNPH1/AdML PPT25, pcDNA3-HNRNPH1/AdML PPT25mt, pcDNA3-HNRNPH1/AdML PPT13, pcDNA3-HNRNPH1/5'AdML, pcDNA3-EML3/AdML PPT25, pcDNA3-EML3/AdML PPT13, and pcDNA3-EML3/5'AdML, were constructed from the parent plasmids by overlap extension PCR with specific primer sets (Supplementary Table S2).

To construct expression plasmids, pcDNA3-Flag-SPF45 and pGEX6p2-SPF45, the ORF region was PCR-amplified from HeLa cells cDNA and subcloned into the pcDNA3-Flag and pGEX6p2 vectors (GE Healthcare Life Sciences). In these plasmids, overlap extension PCR was performed to induce the mutation in the UHM motif of SPF45 (pcDNA3-Flag-SPF45/UHMmt and pGEX6p2-SPF45/UHMmt), to delete the G-patch motif (pcDNA3-Flag-SPF45/ΔG and pGEX6p2-SPF45/ΔG), and to make these siRNA-resistant variants (pcDNA3-Flag-SPF45/siR, pcDNA3-Flag-SPF45/UHMmt/siR and pcDNA3-Flag-SPF45/ΔG/siR).

**Western blotting analyses**. Protein samples were boiled with NuPAGE LDS sample buffer (Thermo Fisher Scientific), separated by SDS-polyacrylamide gel electrophoresis (SDS-PAGE), and the gel was electroblotted onto an Amersham Protran NC Membrane (GE Health Care Life Sciences). The following commercially available antibodies were used to detect targeted proteins: anti-SPF45 (1:1500 dilution, HPA037478, Sigma–Aldrich), anti-SF3b155 (1:3000 dilution; D221-3, MBL Life Science), anti-U2AF65 (1:1500 dilution; U4758, Sigma–Aldrich), anti-U2AF35 (1:1500 dilution; 60289, Proteintech), anti-SF4 (1:1500 dilution; AV40961, Sigma–Aldrich), anti-U1-70K (1:500 dilution; sc-390899, Santa Cruz Biotechnology), anti-GAPDH (1:3000 dilution; M171-3, MBL Life Science), and anti-Flag (1:3000 dilution; anti-DYKDDDDK tag, M185-3L, MBL Life Science). The anti-hPRP43 antibody (1:1500 dilution) was described previously[41]. Immuno-reactive protein bands were detected by the ECL system and visualized by imaging analyzer (ImageQuant LAS 500, GE Healthcare Life Sciences).

**Splicing efficiency screening of siRNA library**. HeLa cells were cultured in Dulbecco's modified Eagle's medium (Wako) supplemented with 10% fetal bovine serum. HeLa cells in 35 mm dishes were transfected with 100 pmol of each siRNA in the Stealth siRNA library targeting 154 human nuclear proteins (Invitrogen–Thermo Fisher Scientific) using Lipofectamine RNAiMax (Invitrogen–Thermo Fisher Scientific) according to the manufacturer's protocol.

At 48–96 h post-transfection, total RNAs were isolated from the siRNA-treated HeLa cells and splicing efficiency was analyzed by RT-PCR using a primer set targeting intron 7 of HNRNPH1 (Supplementary Table S2). The PCR products were separated on 5% PAGE and visualized by imaging analyzer (ImageQuant LAS 500, GE Healthcare Life Sciences). The unspliced pre-mRNA and spliced mRNA were quantified using NIH Image J software, and PSI values were calculated accordingly (Supplementary Table S1). The knockdown efficiencies of all the targeted 154 genes were analyzed by qPCR and ratios to the value of control knockdown were provided (Supplementary Table S1). The sequences of the utilized 308 primers are available upon request.

**siRNA knockdown and splicing assays**. HeLa cells (ATCC) and HEK293 cells (Invitrogen–Thermo Fisher Scientific) cultured in 35 mm dishes were transfected with 100 pmol siRNA using Lipofectamine RNAi max (Invitrogen–Thermo Fisher Scientific) according to the manufacturer's protocol. At 72 h post-transfection, total RNAs were isolated from the siRNA-treated cells using a NucleoSpin RNA kit (Macherey-Nagel). To check depletion of the siRNA-targeted proteins, transfected cells were suspended in Buffer D [20 mM HEPES (pH 7.9), 50 mM KCl, 0.2 mM EDTA, 20% glycerol], sonicated for 20 sec, centrifuged to remove debris, and the lysates were subjected to Western blotting (see above). The siRNAs for SPF45 siRNA#1, SPF45 siRNA#2, U2AF65 siRNA#1[17], hPRP43 siRNA#1 were purchased (Nippon Gene; Supplementary Table S2 for the sequences).

To analyze endogenous splicing products derived from the HNRNPH1, RFC4, EML3, DUSP1, NFKBIA, MUS81, RECQL4, and MTA1 genes, total RNAs from siRNA-treated cells were reverse transcribed by PrimeScript II reverse transcriptase (Takara Bio) with oligo-dT and random primers, and the obtained cDNAs were analyzed by PCR using the specific primer sets (Supplementary Table S2). The PCR products were resolved by 6% PAGE. Splicing products were quantified using NIH Image J software. All the experiments were independently repeated three times and the means and standard errors of the splicing efficiencies were calculated.

To analyze splicing products derived from mini genes, SPF45- and U2AF65-depleted HeLa cells were transfected at 48 h and 68 h post-transfection, respectively, with 0.5 μg of mini-gene plasmid (Supplementary Table S2) using lipofectamine 2000 reagent (Invitrogen–Thermo Fisher Scientific). These cells were incubated for 24 h and 4 h (for Fig. 6 only), respectively, prior to the extraction of RNAs (described above). To analyze splicing products from mini genes, RT-PCR was performed with T7 primer and a specific primer for each mini gene (Supplementary Table S2). All the PCR products were analyzed by 6% PAGE and quantified (described above).

To perform rescue experiments, SPF45-depleted HeLa cells were transfected with 1 μg of pcDNA3-Flag-SPF45/siR, pcDNA3-Flag-SPF45/UHMmt/siR, or pcDNA3-Flag-SPF45/ΔG/siR at 24 h post-transfection. After 48 h culture, total RNA and protein were isolated for RT-PCR and Western blotting, respectively (described above).

In this study, all the oligonucleotide primers were purchased (Fasmac; Supplementary Table S2) and all the PCRs were performed with Blend Taq polymerase (Toyobo Life Science).

**High-throughput RNA sequencing (RNA-Seq) analyses**. Six independent total RNAs derived from HEK293 cells, treated with three control siRNAs and three SPF45-targeted siRNAs, were prepared by NucleoSpin RNA kit (Macherey-Nagel). Then rRNA depletion was performed with the RiboMinus Eukaryote System v2 (Invitrogen–Thermo Fisher Scientific). RNA libraries were prepared using the NEBNext Ultra RNA Library Prep Kit for Illumina (New England Biolabs). These samples were sequenced on the high-throughput sequencing platform (HiSeq2500, Illumina) using a 100 bp single-end strategy.

The sequencing data was analyzed as previously described[42]. Obtained sequence reads were mapped onto the human genome reference sequences (hg19) using the TopHat version 2.1.1 (https://ccb.jhu.edu/software/tophat/index.shtml) and the mapped sequence reads, as BAM files, were assembled using Cufflinks version 2.2.1 (http://cufflinks.cbcb.umd.edu). Using the obtained Cufflinks GTF files as a reference, the BAM files were analyzed using rMATS version 3.2.0 (http://rnaseq-mats.sourceforge.net/)[43] to examine the changes of alternative splicing isoforms. Significant changes of splicing events were defined as when the false discovery rate (FDR) was calculated at less than 0.05.

The strengths of the 5' and 3' splice sites were calculated using MAXENT (http://hollywood.mit.edu/burgelab/maxent/Xmaxentscan_scoreseq.html, http://hollywood.mit.edu/burgelab/maxent/Xmaxentscan_scoreseq_acc.html)[44], and branch point strength, PPT score and PPT length were calculated by SVM-BP finder software (http://regulatorygenomics.upf.edu/Software/SVM_BP/)[45]. The raw data from the RNA-Seq analysis have been deposited in the SRA database (https://www.ncbi.nlm.nih.gov/sra) under accession number GSE135128.

To analyze the sets of retained introns in U2AF65- and SF3b155-depleted HeLa cells, these RNA-Seq data were obtained from the GEO database; the accession numbers are GSE65644 [17] and GSE61603 [46].

**Cellular formaldehyde crosslinking and UV crosslinking followed by immunoprecipitation assays**. To detect Flag-SPF45 association to a pre-mRNA expressed from a reporter mini gene, we performed formaldehyde crosslinking followed by immunoprecipitation as previously described[12]. Briefly, HEK293 cells (in 60 mm dishes) were co-transfected with pcDNA3-Flag-SPF45 and a mini-gene plasmid (pcDNA3-HNRNPH1, pcDNA3-EML3, pcDNA3-MUS81, or pcDNA3-AdML) using Lipofectamine 2000 reagent (Invitrogen–Thermo Fisher Scientific). At 48 h post-transfection, cells were harvested, washed with cold PBS buffer, and fixed with 0.2% formaldehyde for 10 min. The fixation was quenched in 0.15 M glycine (pH 7.0) and cells were washed with PBS. Immunoprecipitations were performed using anti-Flag antibody-conjugated beads to analyze pre-mRNA, from the mini gene, associated with Flag-SPF45.

To detect endogenous U2AF65- and SF3b155-association to a pre-mRNA expressed from a reporter mini gene, we performed UV crosslinking followed by immunoprecipitation as previously described[47,48]. PBS-washed HEK293 cells were

irradiated with 254 nm UV light on ice. The collected cells were lysed and immunoprecipitated with anti-U2AF65 and anti-SF3b155 antibodies.

Immunoprecipitated RNAs were extracted with Trizol reagent (Invitrogen–Thermo Fisher Scientific). The isolated RNAs were reverse transcribed using PrimeScript II reverse transcriptase (Takara Bio) with SP6 primer, and qPCRs were performed using specific primer sets (Supplementary Table S2).

**Biotinylated RNA pull-down assays.** Nuclear extracts were prepared from HEK293 cells transfected with control siRNA or SPF45 siRNA according to the small-scale preparation procedure[49]. Biotin-labeled HNRNPH1 and AdML pre-mRNAs were transcribed with a MEGAscript T7 transcription kit (Invitrogen–Thermo Fisher Scientific) according to the manufacturer's instructions.

The biotinylated pre-mRNA (10 pmol) was immobilized with 5 μL of Dynabeads MyOne StreptavidinT1 magnetic beads (Invitrogen–Thermo Fisher Scientific) according to the manufacturer's instruction. The immobilized pre-mRNA beads were incubated at 30 °C for 15 min in 30 μL reaction mixture containing 30% nuclear extract, RNase inhibitor (Takara Bio) and nuclease-free water. Then NET2 buffer [50 mM Tris (pH 7.5), 150 mM NaCl, 0.05% Nonidet P-40] was added to a final volume of 700 μL and incubated at 4 °C for 1 h. The incubated beads were washed six times with cold NET2 buffer and boiled in SDS-PAGE sample buffer to analyze by Western blotting (described above).

**GST pull-down assays.** GST-SPF45, GST-SPF45/UHMmt, or GST-SPF45/ΔG were expressed in *E. coli* BL21(DE3) CodonPlus (DE3) competent cells (Stratagene–Agilent) and the GST-tagged recombinant proteins were checked by SDS-PAGE followed by Coomassie Blue staining. Induction was carried out at 37 °C for 3 h. The GST-proteins were purified using Glutathione Sepharose 4B (GE Healthcare Life Sciences) according to the manufacturer's protocol.

The recombinant GST-SPF45 proteins (5 μg) were incubated at 30 °C for 15 min in 100 μL mixture containing 30% HeLa cell nuclear extract. After RNase A treatment, NET2 buffer was added to a final volume of 1 mL with 20 μL of Glutathione Sepharose 4B or SF3b155a-antibody conjugated with Protein G Sepharose (GE Healthcare Life Sciences) and incubated at 4 °C for 3 h. The incubated beads were washed six times with cold NET2 buffer and boiled in SDS-PAGE sample buffer to analyze by Western blotting (described above).

**Immunofluorescence microscopic assays.** Immunofluorescence microscopic assays of ectopically expressed Flag-tagged SPF45 proteins were performed as essentially described previously[48].

HeLa cells (in 35 mm dishes) were transfected with 1 μg of pcDNA3-Flag-SPF45/siR, pcDNA3-Flag-SPF45/UHMmt/siR, or pcDNA3-Flag-SPF45/ΔG/siR using lipofectamine 2000 reagent (Invitrogen–Thermo Fisher Scientific). At 48 h post-transfection, cells were fixed with 3% formaldehyde/PBS, permeabilized with 0.1% Triton X-100/PBS, blocked with 5% skimmed milk/PBS and then incubated with the following primary antibodies in 2% skimmed milk/PBS for 0.5 h; anti-SPF45 (1:200 dilution; HPA037478, Sigma–Aldrich), anti-Flag (1:200 dilution; anti-DDDDK tag, M185-3L, MBL Life Sciences). After washing three times with PBS, cells were incubated with Alexa Fluor 488 or Alexa Fluor 568 secondary antibody (Invitrogen–Thermo Fisher Scientific) and then washed five times with PBS. DNA in cells was counter-stained with 4′, 6-diamidino-2-phenylindole (DAPI). The images were analyzed by fluorescence microscope (Olympus).

**Preparation of recombinant proteins.** Recombinant SPF45-G-patch-UHM (234–401) was expressed in pET9d vectors with His6-ProteinA TEV cleavable tag using *E. coli* BL21(DE3) in minimal M9 medium supplemented with [15N]H4Cl for [15N]-labeled protein. Protein expression was induced at $OD_{600}$ around 0.8–1.0 with 1 mM isopropyl-β-D-thiogalactopyranoside (IPTG), followed by overnight expression at 18 °C. Cells were resuspended in 30 mM Tris/HCl (pH 8.0), 500 mM NaCl, 10 mM imidazole with protease inhibitors and lysed using french press. After centrifugation, the cleared lysate was purified with Ni-NTA resin column. The protein sample was further purified by Size-exclusion chromatography on a HiLoad 16/60 Superdex 75 column (GE Healthcare Life Sciences) with 20 mM sodium phosphate (pH 6.5), 150 mM NaCl. The tag was cleaved with TEV protease and removed by Ni-NTA column.

**NMR spectroscopy.** NMR experiments were recorded at 298 K on 500 MHz Bruker Avance NMR spectrometers equipped with cryogenic triple resonance gradient probes. NMR spectra were processed by TOPSPIN3.5 (Bruker), and analyzed using Sparky (T. D. Goddard and D. G. Kneller, SPARKY 3, University of California, San Francisco). Samples were measured at 100 μM protein concentration in the NMR buffer [20 mM sodium phosphate (pH 6.5), 150 mM NaCl, 3 mM DTT] with 10% $D_2O$ added as lock signal. The UHM NMR chemical shift assignment was transferred from the Biological Magnetic Resonance Database (BMRB: 15882). The RNAs [EML3: 5′-GACUGUAUUUGCAGAU-3′, HNRNPH1: 5′-CUCUUGUCCAUCUAGAC-3′] used for the NMR titration were purchased (IBA Lifesciences).

## Data availability

The data supporting the findings of this study are available from the corresponding authors upon reasonable request. The raw data from the RNA-Seq analysis of SPF45-knockdown HeLa cells have been deposited in the Sequence Read Archive (SRA) database under accession number GSE135128. Source data underlying the corresponding figures are provided with this paper.

## Code availability

The Code for the analyses described in this study is available from the corresponding author upon request.

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

## Acknowledgements

We thank Drs. A. R. Krainer, J. Valcárcel, and J. A. Steitz for helpful suggestions and encouragements; Dr. J. Venables for critical reading of the manuscript; H. Shirasaki for technical support, and our lab members for their constructive discussions. K.F. was partly supported by Grants-in-Aid for Scientific Research (C) [Grant number: 18K07304] from the Japan Society for the Promotion of Science (JSPS), a Research Grant from the Hori Sciences and Arts Foundation, and a Research Grant from Nitto Foundation. A.M. was partly supported by Grants-in-Aid for Scientific Research (B) [Grant number: JP16H04705] and for Challenging Exploratory Research [Grant number: JP16K14659] from JSPS. M.S. acknowledges support from the DFG [Grant number: CRC1035, project B03]. The Article Processing Charge for this article was subsidized by Kobe University grant for promoting international joint research.

## Author contributions

K.F. and A.M. conceived and designed the experiments; K.F. performed most of the experiments and analyses, organized the data, and drafted the manuscript; R.Y. performed bioinformatics analyses of the sequencing data; H.-S.K., L.S., and M.S. performed NMR experiments and data analyses; T.H. and K.I. contributed toward establishment and application of the siRNA library for human nuclear proteins; K.F., M.S., and A.M. revised and edited the manuscript. A.M. coordinated and supervised the whole project. All authors read, corrected, and approved the final manuscript.

## Competing interests

The authors declare no competing interests.
