## [Peer Review File · Nature Communications]

REVIEWER COMMENTS

Reviewer #1 (Remarks to the Author):

Fukumura et al report a variety of results consistent with the concept that the splicing factor SPF45 is important for splicing of short introns, which are apparently not dependent on the U2AF heterodimer, thus providing a novel alternative mechanism for 3' splice site recognition in mammalian cells. This conclusion is based upon the special sensitivity of short introns to the RNAi-mediated depletion of SPF45 in cells in culture, which the authors associate, using minigene transfection assays, with the weak polypyrimidine tract typically present in this class of introns. Competition between the UHM domains of SPF45 and U2AF65 for binding to the ULM domain of the U2 snRNP component SF3b155 is proposed as the basis for the differential requirements of SPF45 and U2AF in different intron classes.

These are important results that provide a new paradigm for 3' splice site recognition in higher eukaryotes, as well as an explanation for the old observation that short introns tend to have 3' splice sites harboring weak polypyrimidine tracts. The experiments are generally of high quality and support the conclusions.

In my opinion the following revisions could help to improve the manuscript:

1) The use of the expression "SPF45 is specifically required to splice out short introns" (in Abstract and throughout the text) can be misleading, as it can be interpreted to mean that SPF45 is only involved in the splicing of this class of introns, while results from previous literature as well as the authors' own data indicate that SPF45 is also involved in regulation of splicing of other intron types as well as other classes of splice site switches.

2) I would strongly recommend to provide quantification of results from multiple replicate experiments for the data in Figures 4c (affinity pool down of RNAs followed by western blot analyses) and 6b (competitive binding of U2AF65 and SPF45 to SF3b155), because these results are essential components of the proposed mechanistic model and therefore key for the main conclusions of the work.

3) In the Discussion of the model of SPF45 function, the authors propose that SPF45 is recruited to the 3' splice site through its interaction with SF3b155. It is not clear, however, how this recruitment helps SPF45 to replace U2AF in promoting 3' splice site recognition. U2AF is believed to assist in the recruitment of U2 snRNP by recognizing the polypyrimidine tract / 3' splice site AG and stabilizing the binding of U2 snRNP to the branch point region through its RS domain-dependent targeted annealing between U2 snRNA and the bp as well as through its UHM-ULM interaction with SF3b155. It is not clear how SPF45 could help in U2 snRNP recruitment by "just" interacting with SF3b155 because such interaction could in principle occur in the absence of a pre-mRNA. In other words, it is not clear how the bridging function between pre-mRNA and U2 snRNP achieved by U2AF through its dual RNA and protein binding properties can be achieved by SPF45 solely through its binding to a U2 snRNP protein, and specifically for short introns. I understand that figuring this out may be beyond the scope of this first report, but a more thorough discussion of the model would be helpful.

4) Figure 1: the scheme on the left can be misleading (could be interpreted to mean simultaneous depletion of 154 proteins!). It would be good, in my opinion, to also provide in the scheme a glimpse of the types of nuclear proteins targeted in the screen (categories of hnRNP, SR, RBMs, complex A, etc.).

5) Text:

Abstract : "of the replaced SPF45" is ambiguous and does not convey the idea of competition between UHM-ULM interactions by two factors.

p7: "SPF45 is located exclusively in short introns in fruit fly" is perhaps too strong a conclusion to be

extracted from a limited number of introns analyzed.

p7: "In mammals, SPF45 can cause exon 6 skipping in FAS pre-mRNA that produces soluble isoform of FAS inducing autoimmune phenotypes in mice²²." Ref 22 (Corsini et al) did not address any autoimmunity phenotype in mice.

Reviewer #2 (Remarks to the Author):

In this study, the authors have looked for factors involved in the splicing of short introns. To this aim, they have assessed the impact of the siRNA-mediated depletion of 154 nuclear proteins on the splicing of the 56nt long HNRNPH1's intron 7. This approach revealed that SPF45 (RBM17) is essential for HNRNPH1 intron 7 splicing. By RNAseq, the authors showed that SPF45 depletion results essentially in the significant retention of 187 introns. From those introns, 51 are shorter than 100nt and 39 longer than 1000nt. Yet, when compared to intron retained upon SF3B1 or U2AF65 depletion, introns retained upon SPF45 depletion are enriched with short introns. Further well executed biochemical and in silico analyses, revealed that only the splicing of intron comprising a short "truncated" PPT are dependent on SPF45. On a mechanistic standpoint, the authors convincingly show that SFP45-UHM domain competes with U2AF65 for binding to the SF3b155-ULM motif when the spliceosome assemble on a short intron with truncated PPT.

This manuscript is clearly written and understandable to a large audience. This study is of interest for the RNA and splicing field. However, while technically and conceptually well executed, the conclusions, although interesting, do not represent a major step forward in the understanding of short intron splicing.

Remarks

1-The key question when it comes to short intron splicing is how short introns manage to accommodate the very large spliceosomal machinery and the consequent steric hindrance problems. This point is clearly stated in the introduction "This raises the question of how such ultrashort introns can be recognized and committed to splicing by an 'oversized' A complex without steric hindrance." "Steric hindrance" is not mentioned neither in the result part nor in the discussion. Could the authors explain more clearly how their study help to better understand this important point.

2-The author's goal was to find factors important for short intron splicing. To this aim, the authors have carried out 154 siRNA knockdowns. How have they decided which nuclear protein to test? Why those nuclear protein and not others. While 154 is an important number, yet it is only a fraction of all factors known to take part in splicing and its regulation. The list of factors depleted in this screen is of critical importance as it conditions the possible findings and, as a consequence, the rest of the study. Why not including some core splicing factors like A, B, Bact, C and C* specific proteins? Some of these proteins i.e. Smu1/RED have been shown to have specific function in the splicing of short introns. The authors should comment on this.

3-Moreover, the design of this siRNA screen raises some questions: how the authors have tested for the 154 siRNA depletion efficiencies. This is critical, as some factors may not have affected the splicing of the reporter gene just because of inefficient depletion.

4-In addition, how the authors have differentiated between generic splicing factors i.e. factors required for the splicing of all type of introns, and factors required for the splicing of short intron only. I could not find in their siRNA screen design whether a "long" intron was included as control which would have allowed to assess this. I will take an example to clarify my point. If the splicing of the short and long intron reporters would have been simultaneously inhibited by the siRNA knockdown of a given factor this would have been indicative of a generic splicing defect: not specific of short intron.

Inversely, If only the splicing of the short intron containing reporter transcript would have been affected and not the splicing of the long intron reporter, this would have been indicative of a specific effect on short intron. Could the authors comment on this and explain in more detail their methodology.

5-Regarding this siRNA screening, it would be of great interest for the splicing community, as a resource, to indicate in the list (table S1), for each protein tested, the impact of its depletion on the HNRNPH1 intron 7 splicing. A Percent Spliced In (PSI) would do the trick. Moreover, if the depletion efficiency for each protein has been determined, it would be good to also integrate this value in the Table S1.

6-Figure 3, and second paragraph: "SPF45 is required for splicing on intron with truncated poly-pyrimidine tract (PPT)".

There is something ambiguous about this part as it could be interpreted in two different ways.

First, I would like to agree on definitions: The PPT is a sequence localized between the Branch site (BS) and 3' splice site (3'SS) of introns. However the PPT is not the full sequence that separates the BS from the 3'SS. The PPT is a motif found within that sequence and represents only a fraction of it.

This part shows that intron containing "truncated" short PPT require SPF45 to be spliced. To do this, the authors have replaced the 13nt long PPT of HNRNPH1 intron 7 by the AdML 25nt long PPT. They observed that splicing of HNRNPH1 intron 7 was SPF45 independent when fitted with the AdML PPT. By shortening the AdML PPT to 13nt, splicing of the HNRNPH1 intron 7 was again SPF45 dependent. The authors conclude that PPT length determines whether an intron is spliced in a SPF45 dependent or independent fashion. However, while doing these PPT substitution experiments, they also extended or shortened the distance separating the BS from the 3'SS.

My question is the following: what are we talking about here, the PPT lengths or the BS-3'SS sequence length?

Thus, one alternative Fig 3 interpretation would be: when the sequence separating the BS from the 3'SS is too short ~15nt, intron splicing is SFP45 dependent and this irrespective of the PPT strength or length.

I would strongly urge the authors to clarify this point.

In order to experimentally differentiating between short BS-3'SS sequence and truncated PTT, I would suggest the following: extending the distance separating the BS from the 3'SS of the HNRNPH1 intron 7 and HNRNPH1 intron 7 /AdML 13 PPT constructs. This could be achieved by inserting a 12nt long sequence, which does not contain a PPT, after the end of the existing PPT and the 3'SS. Thus the final BS-3'SS sequence would be 25nt long (like in SPF45 independent introns) but would contain a short 13nt long PPT (like in SPF45 dependent intron). If these constructs would splice normally in a SPF45 independent fashion, this would argue in favor of the BS-3'SS sequence length as determinant for SPF45 requirement in intron splicing. However, if these constructs would not splice in absence of SPF45 i.e. SPF45 dependent, this would clarify the situation and strengthen the author's conclusion that solely the PPT length determines whether a given intron splices in a SPF dependent manner. This point is a major critic.

7-the authors should clearly define what a short intron is. Shorter than 100nt? This point is not defined in the manuscript. Fig S3b suggests that the threshold might be set at 100nt.

8-The RNAseq experiment has allowed to identify 187 introns whose splicing is SPF45 dependent.

Most of these introns are longer than 100 nucleotides and even 39 of them are above 1000 nucleotides in length. For example, XLOC_047811 as referred in the table S2, COL4A5, has a 345 long intron and its splicing is strongly impaired upon SPF45 depletion. Similarly, XLOC_029987 has a 1144 long intron whose splicing is also strongly impaired upon SPF45 depletion. Are those introns also exhibiting short PPTs? Figure S3b seem to indicate that short PPT are only found in introns shorter than 100nt. A rapid examination of XLOC_029987 retained intron suggests that this intron possesses a fairly long PPT. Does that mean that SPF45 depletion impairs the splicing of longer intron through another mechanism independent of the PPT? The authors should comment and clarify this point.

9-In the discussion part: "Our results discover the long-sought-after factor responsible for splicing on short introns". I disagree with this: (i) this is certainly not only one specific factor that helps to splice short introns but very likely several are required (for example, Smu1/RED have already been identified), (ii) based on the result shown here, SPF45 is also very important for the splicing of some long introns (a majority of the intron identified in this study whose splicing is SPF45 dependent are longer than 100nt) (iii) the mechanistic investigation establishes that only the PPT's length matters and not the overall intron length challenging the notion of short introns. Thus, this sentence is clearly misleading.

Reviewer #3 (Remarks to the Author):

In this manuscript, Fukumura K, et al. reveals some human short introns are controlled by SPF45 but not U2AF and proposes a possible mechanism. The authors identified SPF45 as a regulator of a short intron (of *hnrnp1*) via a loss of function screen. RNA-seq of SPF45-depleted cells showed short introns are particularly affected. Using chimera minigenes, the authors showed that the length of PPTs determine SPF45 dependency of splicing. As PPTs are known bound by U2AF65/35, the authors tested and found SPF45 and U2AF compete for binding to intronic RNA and other splicing factors (*sf3b155*). Finally, *sf3b155*-SPF45 interaction is necessary for Spf45's activity of regulating short intron splicing. A conventional thought is that U2AF is ubiquitously required for splicing of all introns. Therefore, this interesting study demonstrates the complex and heterogenous splicing regulation in mammalian cells and shows some short introns depend on SPF45 instead of U2AF. Overall, the study is logical and findings are significant. There are still concerns about interpretation of some results and possible alternative mechanistic models. Data quality can be improved by clarification of various figures (see below) and including quantification (as well as numbers of biological replicates).

Major

1. The manuscript very briefly mentions the siRNA screen and lists the screening targets but does not present the screening result. Is Spf45 the only positive hit? Is it the strongest hit?
2. I like the experiment of swapping the 5'/3' splice sites or the branch site of the *hnrnp1* vs AdML intron, except that the chimera minigenes still contain two of these three elements and that the results are somewhat inconsistent with figs2a. A more rigorous test is a complete swap of these three elements at once to test whether the intron size (or PPT length) is the sole determinant.
3. The authors mention SPF45 does not bind RNA without citation. This is critical to clarify for readers. If direct RNA binding has not been tested, the authors can test SPF45 binding to PPTs of different length using EMSA and/or CLIP.
4. Related to 3, the authors' model suggests *spf45* binding to PPT depends on *p14/SF3b155* binding to branch site. Please provide evidence. A simple test is whether SPF45 mutant still binds to short (vs long) introns in nuclear extract.
5. Fig4a-b, can the authors specify the amplicons they are detecting? Maybe by including a schematic of minigene and labeling the primer positions relative to the minigene.
6. Fig4c, this is a good experiment but requires quantification and statistics. I am also puzzled by the reduced *u2af* binding to the AdML intron upon *spf45* KD. Shouldn't it be enhanced? The same issue for

fig6a-b.

7. In the authors' proposed model, the PPT length to distinct spf45 vs u2af regulation is not consistent with average PPT length of spf45 kd-related introns (figs3b).

8. The authors' model would suggest spf45 does not bind long intron/ppt. But their data suggest otherwise (e.g., fig4a, c). An alternative model is that spf45 bind pervasively to long and short introns, and u2af65 displaces spf45 in long introns. Please compare these two models and discuss the choice of their model.

9. A major defect of spf45 KO in mice is the emergence of cryptic exons (Tan Q et al. 2016). In vivo data is more robust than cell culture data. Do the authors observe cryptic exons in their spf45 depletion? And please discuss the implication of their findings to spf45 regulation of cryptic exons.

10. I think the main conclusion of spf45 regulating intron is assuming that short introns always correlate with short PPTs, which is not always true. I suggest the authors modify their title and conclusion and change short intron to short PPT (or introns with short PPT). What happens to long intron with short PPT, or short introns with a long PPT? are they also regulated by spf45?

Minor

1. Fig S1b, the authors have two siSPF45. which siRNA is used?

2. Fig s3b, the long and short introns are regulated by spf45 differently. The author should discuss about this a bit more. Can the authors include the information about the numbers of introns for each box plot?

3. Is there any reason why fig4a use formaldehyde and fig4b use UV for crosslinking? it is better to use the same crosslinking method consistently.

4. It would be easier for reader if the authors explain in the main text the composition of the biotinylated pre-mRNA (fig4c).

5. There is difference between using cell extracts vs in vitro binding analyses using the purified recombinant proteins which showed that the UHM of SPF45 can bind to the ULMs of SF3b155, U2AF65 and SF1. Please discuss the sources of the differences.

Point-By-Point Responses to Reviewers' Comments

All the changes in the manuscripts are highlighted in RED with the citations in BLUE angle brackets (e.g., <#1-1> indicates the revision according to the comment of Reviewer #1-1)

Reviewer #1:

Fukumura et al. report a variety of results consistent with the concept that the splicing factor SPF45 is important for splicing of short introns, which are apparently not dependent on the U2AF heterodimer, thus providing a novel alternative mechanism for 3' splice site recognition in mammalian cells. This conclusion is based upon the special sensitivity of short introns to the RNAi-mediated depletion of SPF45 in cells in culture, which the authors associate, using minigene transfection assays, with the weak polypyrimidine tract typically present in this class of introns. Competition between the UHM domains of SPF45 and U2AF⁶⁵ for binding to the ULM domain of the U2 snRNP component SF3b155 is proposed as the basis for the differential requirements of SPF45 and U2AF in different intron classes.

These are important results that provide a new paradigm for 3' splice site recognition in higher eukaryotes, as well as an explanation for the old observation that short introns tend to have 3' splice sites harboring weak polypyrimidine tracts. The experiments are generally of high quality and support the conclusions.

In my opinion, the following revisions could help to improve the manuscript:

→ We greatly appreciate for this reviewer's constructive comments, who must be an expert of SPF45 and splicing. We considered all the suggested comments and our manuscript was indeed improved very much.

1. *The use of the expression "SPF45 is specifically required to splice out short introns" (in Abstract and throughout the text) can be misleading, as it can be interpreted to mean that SPF45 is only involved in the splicing of this class of introns, while results from previous literature as well as the authors' own data indicate that SPF45 is also involved in regulation of splicing of other intron types as well as other classes of splice site switches.*

→ We fully agreed and changes the relevant texts accordingly.

2. *I would strongly recommend to provide quantification of results from multiple replicate experiments for the data in Figures 4c (affinity pull-down of RNAs followed by Western blot analyses) and 6b (competitive binding of U2AF⁶⁵ and SPF45 to SF3b155), because these results are essential components of the proposed mechanistic model and therefore key for the main conclusions of the work.*

→ We performed the requested experiments and the quantified data in graphs were added in Figs. 4c and 6b (revised Figs. 5b and 7d). These additional data further confirmed our conclusion; i.e., SPF45 competes out U2AF on truncated poly-pyrimidine tracts (PPTs) and binds to SF3b155 through ULM–UHM interaction.

3. *In the Discussion of the model of SPF45 function, the authors propose that SPF45 is recruited to the 3' splice site through its interaction with SF3b155. It is not clear, however, how this recruitment helps SPF45 to replace U2AF in promoting 3' splice site recognition. U2AF is believed to assist in the recruitment of U2 snRNP by recognizing the polypyrimidine tract / 3' splice site AG and stabilizing the binding of U2 snRNP to the branch point (bp) region through its RS domain-dependent targeted annealing between U2 snRNA and the bp as well as through its UHM-ULM interaction with SF3b155. It is not clear how SPF45 could help in U2 snRNP recruitment by "just" interacting with SF3b155 because such interaction could in principle occur in*

the absence of a pre-mRNA. In other words, it is not clear how the bridging function between pre-mRNA and U2 snRNP achieved by U2AF through its dual RNA and protein binding properties can be achieved by SPF45 solely through its binding to a U2 snRNP protein, and specifically for short introns. I understand that figuring this out may be beyond the scope of this first report, but a more thorough discussion of the model would be helpful.

→ We have not yet elucidated the U2 snRNP recruiting mechanism in the SPF45-dependent/U2AF-independent short introns. The Sharp lab previously reported the U2AF-independent recruitment mechanism that is complemented by SRSF2 (SC35), however, this was a substrate-specific atypical case [MacMillan et al. (1997). *PNAS* 94, 133]. Since we found that G-patch motif in SPF45 is critical for SPF45-dependent splicing, we postulate that the unknown factor interacting SPF45 via G-patch motif helps to recruit U2 snRNP. The identification of such factor is underway. We described this idea in the DISCUSSION.

4. Figure 1: the scheme on the left can be misleading (could be interpreted to mean simultaneous depletion of 154 proteins!). It would be good, in my opinion, to also provide in the scheme a glimpse of the types of nuclear proteins targeted in the screen (categories of hnRNP, SR, RBMs, complex A, etc.).

→ We changed the labeling and added the protein categories in the Fig. 1 (left panel).

5. Text:

(1) Abstract: "of the replaced SPF45" is ambiguous and does not convey the idea of competition between UHM-ULM interactions by two factors.

→ We amended the abstract to avoid the ambiguity.

(2) p7: "SPF45 is located exclusively in short introns in fruit fly" is perhaps too strong a conclusion to be extracted from a limited number of introns analyzed.

→ We agreed and replaced the word 'exclusively' with 'preferentially'.

(3) p7: "In mammals, SPF45 can cause exon 6 skipping in FAS pre-mRNA that produces soluble isoform of FAS inducing autoimmune phenotypes in mice²²." Ref. 22 (Corsini et al) did not address any autoimmunity phenotype in mice.

→ This is right and we added the appropriate citation.

Reviewer #2:

In this study, the authors have looked for factors involved in the splicing of short introns.

To this aim, they have assessed the impact of the siRNA-mediated depletion of 154 nuclear proteins on the splicing of the 56-nt long HNRNPH1's intron 7. This approach revealed that SPF45 (RBM17) is essential for HNRNPH1 intron 7 splicing. By RNA-Seq, the authors showed that SPF45 depletion results essentially in the significant retention of 187 introns. From those introns, 51 are shorter than 100-nt and 39 are longer than 1000-nt. Yet, when compared to intron retained upon SF3B1 or U2AF⁶⁵ depletion, introns retained upon SPF45 depletion are enriched with short introns. Further well executed biochemical and in silico analyses, revealed that only the splicing of intron comprising a short "truncated" PPT are dependent on SPF45. On a mechanistic standpoint, the authors convincingly show that SFP45-UHM domain competes with U2AF⁶⁵ for binding to the SF3b155-ULM motif when the spliceosome assemble on a short intron with truncated PPT.

This manuscript is clearly written and understandable to a large audience. This study is of interest for the RNA and splicing field. However, while technically and conceptually well executed, the conclusions, although interesting, do not represent a major step forward in the understanding of short intron splicing.

→ We are grateful for this reviewer's meticulous reading of the original data and he/she indeed pointed out several critical drawback in our manuscript. We performed all the requested experiments to address these important issues and our manuscript became much more solid.

1. *The key question when it comes to short intron splicing is how short introns manage to accommodate the very large spliceosomal machinery and the consequent steric hindrance problems. This point is clearly stated in the introduction "This raises the question of how such ultrashort introns can be recognized and committed to splicing by an 'oversized' A complex without steric hindrance." "Steric hindrance" is not mentioned neither in the result part nor in the discussion. Could the authors explain more clearly how their study help to better understand this important point.*

→ This study actually could not answer the above question. In fact, the steric hinderance may cause more serious problem in much shorter 'G-rich ultra-short' introns (<49 nt, characterized) (cited Ref. 10, 11). Yet we have no reasonable answer to the question of whether there is steric hinderance problem or not in 56-nt intron that we used for the siRNA screening. To elucidate this problem, we need to analyze spliceosome formation on HNRNPH1 pre-mRNA (56 nt intron) by mass-spec and cryo-electron microscopy. We added such arguments in DISCUSSION.

2. *The author's goal was to find factors important for short intron splicing. To this aim, the authors have carried out 154 siRNA knockdowns. How have they decided which nuclear protein to test? Why those nuclear protein and not others. While 154 is an important number, yet it is only a fraction of all factors known to take part in splicing and its regulation. The list of factors depleted in this screen is of critical importance as it conditions the possible findings and, as a consequence, the rest of the study. Why not including some core splicing factors like A, B, Bact, C and C* specific proteins? Some of these proteins i.e. Smu1/RED have been shown to have specific function in the splicing of short introns. The authors should comment on this.*

→ Honestly we just added/deleted several siRNAs in previously constructed siRNA library that was successfully used for the publication 5 years ago. We arbitrarily selected 154 nuclear proteins including many splicing regulators such as SR proteins, hnRNP proteins, E/A complex proteins, RNA-binding proteins, and so on (now described in revised Fig. 1a). However, we excluded essential splicing factors because we assumed that all splicing must be abolished despite the intron length. The Smu/RED paper (cited Ref. 38 and discussed) was published 3 years after we commenced this study in 2016, therefore we could not include this factor in our siRNA library. We added more mechanistical differences between SPF45 and Smu1/RED in DISCUSSION.

3. *Moreover, the design of this siRNA screen raises some questions: how the authors have tested for the 154 siRNA depletion efficiencies. This is critical, as some factors may not have affected the splicing of the reporter gene just because of inefficient depletion.*

→ We performed qPCR again, using saved cDNA from knockdown cells, to obtain the data of the siRNA depletion efficiencies. The data was shown in the replaced Supplementary Table S1.

4. In addition, how the authors have differentiated between generic splicing factors i.e., factors required for the splicing of all type of introns, and factors required for the splicing of short intron only. I could not find in their siRNA screen design whether a “long” intron was included as control which would have allowed to assess this. I will take an example to clarify my point. If the splicing of the short and long intron reporters would have been simultaneously inhibited by the siRNA knockdown of a given factor this would have been indicative of a generic splicing defect: not specific of short intron. Inversely, If only the splicing of the short intron containing reporter transcript would have been affected and not the splicing of the long intron reporter, this would have been indicative of a specific effect on short intron. Could the authors comment on this and explain in more detail their methodology.

→ Since we excluded siRNAs targeting general or essential splicing factors, we assumed that it was all right to perform control splicing assay after our siRNA screening (as shown in Fig. 2c). To answer rigorously to this reviewer, we performed RT-PCR splicing assay after knockdown the seven representative factors, whose knockdown showed strong splicing repression in the short intron ($PSI > 0.3$ in the revised Table S1). Interestingly, the knockdown of not only SPF45 but also other candidate factors also had no effects at all on pre-mRNA splicing of conventional DUSP1 intron 2 (366 nt). The data were displayed in new Fig. S1a (accordingly original Fig. S1a/b moved to Fig. S1b/c).

5. Regarding this siRNA screening, it would be of great interest for the splicing community, as a resource, to indicate in the list (Table S1), for each protein tested, the impact of its depletion on the HNRNPH1 intron 7 splicing. A Percent Spliced In (PSI) would do the trick. Moreover, if the depletion efficiency for each protein has been determined, it would be good to also integrate this value in the Table S1.

→ We added PSI values (together with the depletion efficiencies) in the revised Table S1.

6. Figure 3, and second paragraph: “SPF45 is required for splicing on intron with truncated poly-pyrimidine tract (PPT)”.

There is something ambiguous about this part as it could be interpreted in two different ways. First, I would like to agree on definitions: The PPT is a sequence localized between the Branch site (BS) and 3' splice site (3'SS) of introns. However the PPT is not the full sequence that separates the BS from the 3'SS. The PPT is a motif found within that sequence and represents only a fraction of it.

This part shows that intron containing “truncated” short PPT require SPF45 to be spliced. To do this, the authors have replaced the 13-nt long PPT of HNRNPH1 intron 7 by the AdML 25-nt long PPT. They observed that splicing of HNRNPH1 intron 7 was SPF45 independent when fitted with the AdML PPT. By shortening the AdML PPT to 13-nt, splicing of the HNRNPH1 intron 7 was again SPF45 dependent. The authors conclude that PPT length determines whether an intron is spliced in a SPF45 dependent or independent fashion. However, while doing these PPT substitution experiments, they also extended or shortened the distance separating the BS from the 3'SS.

My question is the following: what are we talking about here, the PPT lengths or the BS-3'SS sequence length? Thus, one alternative Fig 3 interpretation would be: when the sequence separating the BS from the 3'SS is too short ~15-nt, intron splicing is SFP45 dependent and this irrespective of the PPT strength or length. I would strongly urge the authors to clarify this point.

In order to experimentally differentiating between short BS-3'SS sequence and truncated PTT, I would suggest the following: extending the distance separating the BS from the 3'SS of the HNRNPH1 intron 7 and HNRNPH1 intron 7/AdML 13 PPT constructs. This could be achieved by inserting a 12-nt long sequence, which does not contain a PPT, after the end of the existing PPT and the 3'SS. Thus the final BS-3'SS sequence would be 25-nt long (like in SPF45 independent introns) but would contain a short 13-nt long PPT (like in SPF45 dependent intron). If these

constructs would splice normally in a SPF45-independent fashion, this would argue in favor of the BS-3'SS sequence length as determinant for SPF45 requirement in intron splicing. However, if these constructs would not splice in absence of SPF45, i.e. SPF45-dependent, this would clarify the situation and strengthen the author's conclusion that solely the PPT length determines whether a given intron splices in a SPF dependent manner. This point is a major critic.

→ To answer the above critical questions, we made additional two mini-gene constructs; i.e., HNRNPH1 intron 7/+12nt and HNRNPH1 intron7/AdML PPT13/+12nt (we inserted two copies of XhoI site to extend 12 nt between BS and 3'SS). *In cellulo* splicing showed SPF45-dependent splicing was not changed even extending the distance between BS and 3'SS (showed in new Fig. S3a). Therefore, we could keep our original conclusion; namely the length of PPT is the determinant for SPF45-dependent splicing. The explanation (with the citation of Fig S3a) was added in the text accordingly.

7. The authors should clearly define what a short intron is. Shorter than 100-nt? This point is not defined in the manuscript. Fig S3b suggests that the threshold might be set at 100-nt.

→ We agreed and we defined introns shorter than 100 nt as 'short introns' in this study. We explained in the text just after citation of Fig. S4 (original Fig. S3b).

8. The RNA-Seq experiment has allowed to identify 187 introns whose splicing is SPF45 dependent.

Most of these introns are longer than 100 nucleotides and even 39 of them are above 1000 nucleotides in length. For example, XLOC_047811 as referred in the Table S2, COL4A5, has a 345-nt long intron and its splicing is strongly impaired upon SPF45 depletion. Similarly, XLOC_029987 has a 1144-nt long intron whose splicing is also strongly impaired upon SPF45 depletion. Are those introns also exhibiting short PPTs? Figure S3b seem to indicate that short PPT are only found in introns shorter than 100-nt. A rapid examination of XLOC_029987 (*mistaken for XLOC_047811?*) retained intron suggests that this intron possesses a fairly long PPT. Does that mean that SPF45 depletion impairs the splicing of longer intron through another mechanism independent of the PPT? The authors should comment and clarify this point.

→ Thanks for the interesting question. The answer is YES. We checked the PPT lengths of the requested two genes (XLOC_047811, XLOC_029987) and summarized as follows.

Gene ID	Intron length (nt)	PPT length (nt)	PPT score
XLOC_047811	345	18	40
XLOC_029987	1144	10	23

The 1144-nt intron has a typical truncated PPT with low PPT score, whereas the 345-nt intron has longer PPT with higher PPT score, which appears the standard PPT (Cf. Fig. 3 for these values, see Fig. S14 of cited Ref. 44). We acknowledge that a subset of the SPF45-dependent long introns (>100 nt) do not necessarily possess truncated PPTs. Therefore, we briefly mentioned in the legend of original Fig. S3b (now Fig. S4); i.e., there is another unknown mechanism in SPF45-dependent splicing with a standard PPT. This fact was not surprising since it was previously shown that SPF45 interacts with U2SURP and CHERP and regulates various alternative splicing including intron-retention (cited Ref. 37). Nevertheless, we could conclude that the set of SPF45-dependent short introns (<100 nt) possess the significantly shorter PPTs (statistically analyzed in original Fig. S3b, now Fig. S4), which are spliced out by our proposed mechanism. The fact that the knockdown of U2SURP (SR140) has no effect on splicing of HNRNPH1 with 56-nt intron (Table S1) further supports this idea. We added these considerable arguments in DISCUSSION. The tile was also modified to be more accurate.

9. *In the discussion part: “Our results discover the long-sought-after factor responsible for splicing on short introns”. I disagree with this: (i) this is certainly not only one specific factor that helps to splice short introns but very likely several are required (for example, Smu1/RED have already been identified), (ii) based on the result shown here, SPF45 is also very important for the splicing of some long introns (a majority of the intron identified in this study whose splicing is SPF45-dependent are longer than 100-nt) (iii) the mechanistic investigation establishes that only the PPT’s length matters and not the overall intron length challenging the notion of short introns. Thus, this sentence is clearly misleading.*

→ We deleted this problematic sentences. We fully agreed with this reviewer’s point and we reflected such aspect in DISCUSSION (see also our answer in 8).

Reviewer #3:

In this manuscript, Fukumura K, et al. reveals some human short introns are controlled by SPF45 but not U2AF and proposes a possible mechanism. The authors identified SPF45 as a regulator of a short intron (of HNRNPH1) via a loss of function screen. RNA-Seq of SPF45-depleted cells showed short introns are particularly affected. Using chimera minigenes, the authors showed that the length of PPTs determine SPF45-dependency of splicing. As PPTs are known bound by U2AF^{65/35}, the authors tested and found SPF45 and U2AF compete for binding to intronic RNA and other splicing factors (SF3b155). Finally, SF3b155-SPF45 interaction is necessary for SPF45’s activity of regulating short intron splicing. A conventional thought is that U2AF is ubiquitously required for splicing of all introns. Therefore, this interesting study demonstrates the complex and heterogenous splicing regulation in mammalian cells and shows some short introns depend on SPF45 instead of U2AF.

Overall, the study is logical and findings are significant. There are still concerns about interpretation of some results and possible alternative mechanistic models. Data quality can be improved by clarification of various figures (see below) and including quantification (as well as numbers of biological replicates).

→ We greatly appreciate for this reviewer’s positive evaluation. All the valuable comments were considered and our manuscript was amended accordingly.

Major Remarks

1. *The manuscript very briefly mentions the siRNA screen and lists the screening targets but does not present the screening result. Is SPF45 the only positive hit? Is it the strongest hit?*

→ Yes, SPF45 was the strongest hit, but not the only positive hit. We performed qPCR again using the saved cDNA and obtained the PSI values and the depletion efficiencies, which were added in the revised Table S1 (see also Reviewer #2-3, 5).

2. *I like the experiment of swapping the 5'/3' splice sites or the branch site of the HNRNPH1 vs AdML intron, except that the chimera minigenes still contain two of these three elements and that the results are somewhat inconsistent with Fig 2a. A more rigorous test is a complete swap of these three elements at once to test whether the intron size (or PPT length) is the sole determinant.*

→ We performed the requested rigorous test with one mini-gene construct, HNRNPH1 intron 7/AdML 5’SS-BP-3’SS. *In cellulo* splicing showed evident SPF45-dependency (displayed in the additional panel of Fig. S2b). The data are sufficient to keep our original conclusion; namely the truncated PPT is the solo determinant for SPF45-dependent splicing.

3. *The authors mention SPF45 does not bind RNA without citation. This is critical to clarify for readers. If direct RNA binding has not been tested, the authors can test SPF45 binding to PPTs of different length using EMSA and/or CLIP.*

→ **Previously it was shown that mammalian SPF45 did not bind directly to RNA *in vitro* (Ref. 22 was cited). Therefore we collaborated with the Sattler lab, and we confirmed this fact by NMR data (original Fig. S7, now Fig. S9). Therefore, we used formaldehyde crosslinking (cited Ref. 12) to detect any indirect RNA association of SPF45.**

4. *Related to 3, the authors' model suggests SPF45 binding to PPT depends on p14/SF3b155 binding to branch site. Please provide evidence. A simple test is whether SPF45 mutant still binds to short (vs long) introns in nuclear extract.*

→ **We performed the formaldehyde crosslinking assay to detect indirect SPF45 (wild-type and mutants) binding to PPT of three kinds of SPF45-dependent pre-mRNAs (displayed in new Fig. S8). The marked impairment of this binding with SPF45-UHM mutant implicated that the indirect SPF45 binding to PPT is dependent on SF3b155 binding through ULM-UHM interaction (see model original Fig. 8 now Fig. 9). We added relevant descriptions in the RESULTS.**

5. *Fig. 4a-b, can the authors specify the amplicons they are detecting? Maybe by including a schematic of minigene and labeling the primer positions relative to the minigene.*

→ **We added the schematic mini-gene structures with utilized primers below the Fig. 4a/b.**

6. *Fig. 4c, this is a good experiment but requires quantification and statistics. I am also puzzled by the reduced U2AF binding to the AdML intron upon SPF45 KD. Shouldn't it be enhanced? The same issue for Fig. 6a,b.*

→ **We added the quantified graphs in Figs. 4c and 6b (revised Figs. 5b and 7d; same request by Reviewer #1-2). Regarding the slightly reduced U2AF⁶⁵ binding to the AdML intron upon the SPF45 knockdown could be due to the varied amounts of the SF3B155 and U2AF⁶⁵ prepared from different knockout extracts. Importantly, such difference was not significant by the statistical analysis (see error bars of the revised Fig. 5b-right panel).**

7. *In the authors' proposed model, the PPT length to distinct SPF45 vs U2AF regulation is not consistent with average PPT length of SPF45 KD-retained introns (Fig. S3b).*

→ **In the schematic model figure (Fig. 8), we did not draw the PPT length with 'U' characters in scale. We changed the misleading drawing in Fig. 8 (revised Fig. 9).**

8. *The authors' model would suggest SPF45 does not bind long intron/PPT. But their data suggest otherwise (e.g., Fig. 4a, c). An alternative model is that SPF45 bind pervasively to long and short introns, and U2AF⁶⁵ displaces SPF45 in long introns. Please compare these two models and discuss the choice of their model.*

→ **Even in long intron, SPF45 and U2AF⁶⁵ associates with the U2 snRNP simultaneously, through the binding to SF3b155, because SF3b155 has five ULMs. But the associated SPF45 is not functional, but U2AF⁶⁵ is functional, to splice out long intron. We had explained this fact in DISCUSSION.**

9. *A major defect of SPF45 KO in mice is the emergence of cryptic exons (Tan Q et al. 2016, cited*

Ref. 26). *In vivo* data are more robust than cell culture data. Do the authors observe cryptic exons in their SPF45 depletion? And please discuss the implication of their findings to SPF45 regulation of cryptic exons.

→ We respect the previous SPF45-knockout study *in vivo* (cited Ref. 26) and SPF45-knockdown study *in cellulo* (cited Ref. 37) very much. As this reviewer suggested, we investigated the unannotated cryptic exons with our RNA-Seq data of the SPF45 knockdown HEK293 cell extracts using the 'Crypsplice' tool (cited Ref. 26). We used the same conditions in Ref. 26, and found the numbers of cryptic exons gains and losses were 59 and 123, respectively, some of which may reflect SPF45-dependent splicing in short introns. We mentioned this idea in DISCUSSION.

In cited Ref. 37, the knockdown of SPF45, U2SURP or CHERP in HEK293 cells caused the changes of the cryptic exons and showed considerable overlap across the three factors, implicating that the interaction of SPF45 with U2SURP or CHERP plays a role in regulating the usage of cryptic exons. However, our siRNA screening demonstrated that the knockdown of U2SURP had no effect on the SPF45-dependent splicing of short introns, indicating that the mechanism of SPF45 as a regulator of alternative exon usage is distinct from that of the SPF45-dependent splicing of short intron. Remarkably, this fact is also supported by the data in Fig. 6A of cited Ref. 37; i.e., the observed 4 events of retained-intron inclusion (that must include the repression of short introns) by the SPF45 knockdown is not overlapped (0 event) by knockdown of U2SURP, or CHERP. We added the above findings in DISCUSSION.

10. *I think the main conclusion of SPF45 regulating intron is assuming that short introns always correlate with short PPTs, which is not always true. I suggest the authors modify their title and conclusion and change short intron to short PPT (or introns with short PPT). What happens to long intron with short PPT, or short introns with a long PPT? Are they also regulated by SPF45?*

→ This comments are essentially same as in Reviewer #2-8. First of all, we changed the title appropriately. We noticed that there is a subset of the SPF45-dependent long introns (>100 nt) that do not necessarily possess truncated PPTs, whose splicing mechanism is not identified yet (see our explanation in the legend of Fig.3; see also Reviewer #2-8 for the typical examples). However, we could demonstrate that the set of SPF45-dependent short introns (<100 nt) possess the significantly shorter PPTs (statistical analysis in original Fig. S3b, now Fig. S4), which are spliced out by our proposed mechanism. We added these rigorous explanations in DISCUSSION.

Minor Remarks

11. *Fig S1b, the authors have two siSPF45. which siRNA is used?*

→ Thanks. We used the siRNA 'SPF45 #2' (Fig. S1a, now Fig. S1b) in the knockdown experiments in Fig. S1b (now Fig. S1c) and Fig. 1b. We described this in the legend of the Fig. S1c. and Fig. 1b

12. *Fig S3b, the long and short introns are regulated by SPF45 differently. The author should discuss about this a bit more. Can the authors include the information about the numbers of introns for each box plot?*

→ As described in Reviewer #2-8, previous report showed that SPF45 interacts with U2SURP and CHERP and regulates various alternative splicing including intron-retention (cited Ref. 37). We speculate that such mechanism would be involved in the SPF45-dependent intron with long PPT, which is under investigation. We added our idea in DISCUSSION. As

suggested, the numbers of introns were added in the legend of Fig S4 (original Fig. S3b). We also added the numbers of introns in Fig. 2b.

13. *Is there any reason why Fig. 4a use formaldehyde and Fig. 4b use UV for crosslinking? it is better to use the same crosslinking method consistently.*

→ Because mammalian SPF45 cannot bind directly to RNA *in vitro* (cited Ref. 22), we used formaldehyde crosslinking to detect any indirect RNA association of SPF45 (cited Ref. 12). The answer was described in the text.

14. *It would be easier for reader if the authors explain in the main text the composition of the biotinylated pre-mRNA (Fig. 4c).*

→ We added the brief explanation in the main text (for now Fig. 5a) to be easier for readers.

15. *There is difference between using cell extracts vs in vitro binding analyses using the purified recombinant proteins which showed that the UHM of SPF45 can bind to the ULMs of SF3b155, U2AF⁶⁵ and SF1. Please discuss the sources of the differences.*

→ Previously, the *in vitro* binding between SPF45-UHM and SF3b155-ULM or U2AF⁶⁵-UHM and SF3b155-ULM was shown with the partial recombinant proteins including the corresponding domains (cited Ref. 22). While in our experiments, we used the full-length recombinant proteins and the binding assays were performed with crude cell extracts, which allows more physiological interactions with other proteins. We took advantage of such *in cellulo* assays to demonstrate functional interactions, which were not obtained by *in vitro* assays; i.e., SPF45-UHM binding to SF3b155-ULM but not to SF1-ULM and U2AF⁶⁵-ULM. These observation allowed us to propose the model that the SPF45-UHM and U2AF⁶⁵-UHM compete for a functional binding toward the SF3b155-ULM (see Fig. 9). We revised the explanation to make it clear in RESULTS.

REVIEWERS' COMMENTS

Reviewer #1 (Remarks to the Author):

The authors have adequately addressed the issues raised in my previous report and therefore I fully support publication of the manuscript in Nature Communications.

Juan Valcarcel

Reviewer #2 (Remarks to the Author):

I would like to thank the authors for having addressed all points raised by the referees.

In my view, the new data added to the initial dataset clarify all critical points and strengthen the author's conclusions.

While I believe this manuscript is now suitable for publication, I would like to make few additional remarks. The authors might be willing to consider them.

1 Sup table 1: PSI upon all Knockdowns are indicated and very helpful. However, I may have missed it, what is the PSI upon wild type situation or mock siRNA depletion? Or, are the PSI shown here DeltaPSI? If these are not deltaPSI, I would recommend to add the Control depletion's PSI to the list. This would greatly help the reader to evaluate how the different knockdowns alter HNRNPH1 intron 7 splicing.

2 Fig 6, Panel B: Surprisingly, in control cells, EML3 intron 70 and MUS81 intron 13 appear to be hardly spliced ; the vast majority is unspliced with very little signal corresponding to the spliced form. In all other figures, e.g. Fig1b, Fig2c, FigS1b,c, EML3 intron 70 and MUS81 intron 13 are almost entirely spliced in control cells.

This is quiet puzzling to me.

Although the conclusion derived from this figure are in agreement with the data presented, I am wondering whether similar effect upon U2AF65 knockdown would be observed if EML intron 70 and MUS81 intron 13 would splice as efficiently in the control condition as shown in the rest of the study.

I will try to clarify my point: it is stated that U2AF Knockdown favors splicing of SPF35 dependent introns. This conclusion derives from the comparison between splicing efficiencies observed in control condition and upon U2AF65 knockdown. In fig. 6b, splicing of the reporter introns appear more efficient upon U2AF65 knockdown. However, the reporter introns appear better spliced essentially because, all of a sudden, they do not splice anymore in the control condition. Consequently, the reporter introns appear better spliced upon U2AF65 Knockdown although such conclusion could not be made if the reporter introns would have spliced normally in the control.

In my view, this point challenges the conclusion derived from this figure. It might need some clarifications as to why EML intron 70 and MUS81 intron 13 do not splice in control cells.

3 Splicing of 187 introns appear to be SPF45 dependent. This dependency relies on truncated PPTs. It would be interesting to indicate how many introns within the whole genome have truncated PPTs and whose PPT's score predict that their splicing would be SPF45 dependent. This information would be of great value to evaluate the extent of SPF45 role in short intron splicing.

4 In the discussion: "Our considerable finding is that U2AF65 needs to be expelled by SPF45 to promote splicing of a subset of short introns".

This is a very interesting point. However, this point might need to be discussed further. Indeed, this also suggests that, if not expelled and stays associated with its RNA substrate, U2AF65 may prevent the efficient splicing of short introns with truncated PPTs (supported by the results presented fig. 5a and 6b). In this scenario, the mere U2AF65 destabilization might be sufficient to activate short intron splicing. Thus, SPF45 role would solely consists in displacing U2AF65. Consequently, the short introns described in this study might be efficiently spliced in cells lacking both U2AF65 and SPF45. It may be very interesting to, at least, briefly discuss this point.

5 Few sentences in this manuscript appear slightly odd as if some words would be missing. This may be necessary to check this.

Reviewer #3 (Remarks to the Author):

The authors have addressed my concerns except point 8. I could not find clear explanation in results or discussion to dismiss the alternative model. This is an important point for the mechanistic model, so I feel the authors can and should explain it better, which will demonstrate the authors' insights and help readers better understand the logic of deriving this mechanism.

Point-By-Point Responses to Reviewers' Second Comments

All the changes in the manuscripts are highlighted in BLUE.

Reviewer #1:

The authors have adequately addressed the issues raised in my previous report and therefore I fully support publication of the manuscript in Nature Communications. Juan Valcarcel

→ We are very glad to hear the support of our new findings of SPF45 from an pioneer of SPF45 research, Dr. Valcarcel. Thank you very much.

Reviewer #2:

In my view, the new data added to the initial dataset clarify all critical points and strengthen the author's conclusions.

While I believe this manuscript is now suitable for publication, I would like to make few additional remarks. The authors might be willing to consider them.

→ Thanks for your support toward the publication. Of course, we are willing to address your following additional remarks.

1. Sup. Table S1: PSI upon all Knockdowns are indicated and very helpful. However, I may have missed it, what is the PSI upon wild-type situation or mock siRNA depletion? Or, are the PSI shown here DeltaPSI? If these are not deltaPSI, I would recommend to add the Control depletion's PSI to the list. This would greatly help the reader to evaluate how the different knockdowns alter HNRNPH1 intron 7 splicing.

→ The PSI values added in Table S1 are not DeltaPSI values. We used a negative control siRNA for a control depletion, and we added this control value (PSI=0.103) in Table S1. We also added the definition of PSI (Percent Spliced in) in this case (Percent Retained-intron In) in the margin of Table S1.

2. Fig. 6, Panel B: Surprisingly, in control cells, EML3 intron 17 and MUS81 intron 13 appear to be hardly spliced; the vast majority is unspliced with very little signal corresponding to the spliced form. In all other figures, e.g. Fig. 1b, Fig. 2c, Fig. S1b,c, EML3 intron 17 and MUS81 intron 13 are almost entirely spliced in control cells. This is quiet puzzling to me.

Although the conclusion derived from this figure are in agreement with the data presented, I am wondering whether similar effect upon U2AF⁶⁵ knockdown would be observed if EML intron 70 and MUS81 intron 13 would splice as efficiently in the control condition as shown in the rest of the study.

I will try to clarify my point: it is stated that U2AF Knockdown favors splicing of SPF35 dependent introns. This conclusion derives from the comparison between splicing efficiencies observed in control condition and upon U2AF⁶⁵ knockdown. In Fig. 6b, splicing of the reporter introns appear more efficient upon U2AF⁶⁵ knockdown. However, the reporter introns appear better spliced essentially because, all of a sudden, they do not splice anymore in the control condition. Consequently, the reporter introns appear better spliced upon U2AF⁶⁵ Knockdown although such conclusion could not be made if the reporter introns would have spliced normally in the control.

In my view, this point challenges the conclusion derived from this figure. It might need some clarifications as to why EML intron 70 and MUS81 intron 13 do not splice in control cells.

→ As we mentioned in the manuscript, the former experiments (Fig. 6) are *in cellulo* splicing of transfected reporter mini-genes, whereas the latter experiments (Figs. 1b, 2c, S1b/c) are detection of endogenous splicing. It is reasonable to see much worth splicing efficiency with over-expressed pre-mRNA, excess of the substrates, in the former case. To observe

the maximum splicing stimulation effects under U2AF⁶⁵-knockdown, we have to use much shorter incubation time (4 h instead of 24 h; see Materials and Methods), which is the reason why apparent splicing efficiency in Fig. 6 is poor. We briefly mentioned this technical reason in the legend of Fig. 6.

3. Splicing of 187 introns appear to be SPF45 dependent. This dependency relies on truncated PPTs. It would be interesting to indicate how many introns within the whole genome have truncated PPTs and whose PPT's score predict that their splicing would be SPF45 dependent. This information would be of great value to evaluate the extent of SPF45 role in short intron splicing.

→ It is also intriguing questions for us. In practice, it is currently impossible to verify the SPF45-dependency of all introns with truncated PPT in the genome by analytical methods such as RNA-Seq. We are planning to dissolve this demanding problem.

4. In the discussion: "Our considerable finding is that U2AF⁶⁵ needs to be expelled by SPF45 to promote splicing of a subset of short introns".

This is a very interesting point. However, this point might need to be discussed further. Indeed, this also suggests that, if not expelled and stays associated with its RNA substrate, U2AF⁶⁵ may prevent the efficient splicing of short introns with truncated PPTs (supported by the results presented in Fig. 5a and 6b). In this scenario, the mere U2AF⁶⁵ destabilization might be sufficient to activate short intron splicing. Thus, SPF45 role would solely consists in displacing U2AF⁶⁵. Consequently, the short introns described in this study might be efficiently spliced in cells lacking both U2AF⁶⁵ and SPF45.

It may be very interesting to, at least, briefly discuss this point.

→ Your idea is logical, however, we predict the additional role of SPF45 to recognize the 3' splice site. Since our NMR study showed that SPF45 cannot bind RNA directly (Fig. S9), it is reasonable to assume there is a SPF45-interacting factor, such as U2AF⁶⁵-interacting U2AF³⁵. Our further study to elucidate the recognition mechanism of the 3' splice site is underway. We described this argument in DISCUSSION. Thanks.

5. Few sentences in this manuscript appear slightly odd as if some words would be missing. This may be necessary to check this.

→ We carefully checked the manuscript. We are sorry but we could not find these few sentences that were not pointed out explicitly.

Reviewer #3:

The authors have addressed my concerns except point 8. I could not find clear explanation in results or discussion to dismiss the alternative model. This is an important point for the mechanistic model, so I feel the authors can and should explain it better, which will demonstrate the authors' insights and help readers better understand the logic of deriving this mechanism.

→ We are sorry that we forgot to add the clear explanation in the text. Previous report [Corsini *et al.* (2007). *Nat. Struct. Mol. Biol.* 14, 620] and our NMR study (Fig. S9) indicated that SPF45 cannot bind RNA directly. We showed SPF45 binding to both short and long introns (Fig. 4a), but that is due to protein-protein interaction between SPF45-UHM and SF3b155-ULM (as SF3b155 has 5 ULMs). Therefore, both U2AF⁶⁵ and SPF45 can bind to long introns simultaneously, and thus, U2AF⁶⁵ does not displace SPF45 in long introns. Indeed, both SPF45 and U2AF⁶⁵ are detected in spliceosomal A complex from the results of mass spectrometric studies using long intron substrates [reviewed in Wahl *et al.* (2015). *Cell* 161, 1474.]. Together, your suggested alternative model, "SPF45 bind pervasively to long and short introns and U2AF⁶⁵ displaces SPF45 in long introns", is unlikely. Now, we added this explanation in the legend of Figure 9.